# Comparison of machine learning methods in forecasting and characterizing the birch and grass pollen season

Daniel Bulanda [1]*, Małgorzata Bulanda[2], Małgorzata Sacha[2], Adrian Horzyk[1], Dorota Myszkowska[2]

**1** Department of Biocybernetics and Biomedical Engineering, AGH University of Krakow, Krakow, Malopolska, Poland, **2** Department of Clinical and Environmental Allergology, Jagiellonian University Medical College, Krakow, Malopolska, Poland

\* daniel@bulanda.net

## Abstract

The primary approach to the treatment of seasonal allergic diseases involves minimizing exposure to allergens and initiating early personalized therapy. The medication should be introduced about 7 days before the start of the pollen season and intensified during the period of the highest concentrations of sensitizing pollen. Therefore, forecasts for the concentration of pollen that causes clinical symptoms are of indisputable value to both doctors and patients. The study was carried out in Krakow (Southern Poland) with birch (*Betula*) and grasses (Poaceae) pollen data collected using the volumetric method in 1991-2024. The following meteorological data were collected and used in the study: temperature (mean, minimum and maximum), humidity, cloud cover, sunshine duration, mean wind speed, mean pressure at sea level, global radiation and snow depth. Eight machine learning models from four distinct families (lazy, linear, tree-based, and deep learning) were chosen to estimate the probability of the occurrence of pollen concentration within specified categories. These predictions were based on meteorological data combined with pollen concentration levels in the preceding days. Using the occurrence of pollen concentration in the selected categories as the target variable, the top-performing models achieved accuracies of 92.2%, 88.3%, and 87.2% for 1-day, 4-day, and 7-day forecasts of *Betula* pollen, respectively. Similarly, for Poaceae pollen, the models achieved 86.1%, 81.8%, and 80.0% accuracy for predictions of 1 day, 4 days, and 7 days ahead, respectively. In addition, a feature importance analysis and an association rule mining were performed to explain the dependencies between pollen concentration and meteorological variables. The tested machine learning methods achieve results that allow for satisfactory efficiency in predicting pollen concentration for up to seven consecutive days. The best-performing machine learning methods

**Data availability statement:** The meteorological data files are available from the European Climate Assessment & Dataset project database: https://www.ecad.eu/dailydata/predefinedseries.php. To provide a minimal data set necessary to replicate our analyses, two csv files with Betula pollen and two files with Poaceae pollen data obtained in 2022 and 2023, and the meteorological data of ten selected factors were uploaded to the Open Science Framework with DOI number: DOI 10.17605/OSF.IO/9YZCF (https://osf.io/9yzcf). Requests for access to the whole dataset underlying this study should be directed to the Department of Clinical and Environmental Allergology Jagiellonian University Medical College via email: zaklad.alergologii@cm-uj.krakow.pl. Researchers seeking access must meet the criteria for access to confidential data, this means providing a research plan and confirming compliance with data use and confidentiality policies, i.e. use them to a given, specific study aim/purpose and not sharing with a third party.

**Funding:** The study was supported by the statutory project of the Ministry of Science and Higher Education in Poland N41/DBS/001323. Initials of the authors who received the award: MB. URL of the funder: https://www.gov.pl/web/science. The funders had no role in study design, data collection and analysis, decision to publish, or preparation of the manuscript.

**Competing interests:** The authors have declared that no competing interests exist.

were boosted trees, associative knowledge graphs, and deep neural networks with memory cells.

## Introduction

According to epidemiological studies, the current prevalence of allergic rhinitis (AR) in the world, depending on geographical latitude, is 5-50%. The incidence of pollen-induced allergic asthma worldwide ranges from 1% to 18% of the population. [1–4]. In recent years, we have observed an increase in the incidence of allergic atopic diseases in the world [5,6], which may be related to the changing environmental factors that influence the seasons of plant pollen [5].

The prediction and early within-season characterization of pollen dynamics are of great importance for patients with pollen allergies and medical doctors. Seasonal AR negatively impacts overall quality of life, causes absences from work and school, and generates enormous costs to the healthcare system. Untreated AR increases the risk of poor asthma control and exacerbations [7]. The basic treatment for seasonal allergic diseases is to avoid exposure to the allergen and early and individually selected therapy, including allergen specific immunotherapy [8]. The medication to control symptoms should be introduced approximately 7 days before the pollen season starts and intensified during the highest concentrations of sensitizing pollen [9].

Modern machine learning methods can help characterize pollen seasons, determine which environmental factors influence them strongly, and most importantly, predict the changes in pollen concentration during pollen seasons [10,11]. In such scenarios, commonly used methods include linear models, boosted trees, and deep neural networks [10,12–15].

Traditionally, aerobiological time-series have been modeled using statistical approaches such as multiple linear regression, generalized additive models (GAMs), ARIMA, and dynamic regression [12,16,17]. While these methods are straightforward and interpretable, they rely on assumptions of linearity, normality, and homoscedasticity, and may struggle to capture complex, non-linear interactions between meteorological drivers and pollen release dynamics.

In recent years, machine learning techniques have emerged as powerful alternatives, capable of flexibly modeling non-linear relationships and high-order interactions without strict distributional assumptions. Astray et al. [18] developed Random Forest, Support Vector Machine, and neural network models on 24 years of *Parietaria* pollen data in northwest Spain, achieving a mean absolute error of approximately one day in peak-date prediction and RMSE 5.55 - 5.84 for one day ahead prediction. Cordero et al. [19] applied LightGBM and neural network ensembles to 20 years of Olea pollen data, accurately forecasting season-peak timing (mean peak date error < 1 day) and daily concentrations (RMSE 25.03 - 29.27). In Switzerland, Shokouhi et al. [20] compared linear models (LASSO, Ridge and Elastic net), nonlinear models (XGBoost, Random forest and neural networks) and ensembling approaches for birch and grass pollen, finding tree-based and hybrid models outperform linear approaches. For the single-model approach they achieved RMSE 23.0 - 27.7 for grass and RMSE

105.7 - 140.7 for birch. Zewdie et al. [21] similarly demonstrated that Random Forests, Support Vector Machines, and neural networks could predict *Ambrosia* pollen with $R^2$ between 0.21 and 0.37, with Random Forest producing superior performance to the other models tested.

Despite these promising results, few studies have systematically benchmarked a broad suite of both linear and non-linear algorithms on long-term pollen datasets spanning multiple taxa. The aim of our work was to select the optimal machine learning methods for the prediction and characterization of pollen seasons. Here, we focus on several popular and representative models that have shown their effectiveness in pollen seasons forecasting. Our goal is to reveal their strengths and weaknesses, as well as application scenarios.

## Materials and methods

### Data

Study was performed in Krakow (Southern Poland; near the grid point 50°$N$ -20°$E$). The city is surrounded by farmland and forests that prevail west of the city. The study area corresponds to the municipality of Krakow, which in 2021 covered an area of 327 km$^2$ and had a population of 780 796 inhabitants [22]. Krakow is located in a moderate, warm, and transitional climate between maritime and continental air masses.

Daily pollen concentrations were obtained within the framework of regular airborne pollen analyzes performed by the Aerobiological Monitoring Station at the Department of Clinical and Environmental Allergology of the Jagiellonian University Medical College in Krakow for 34 years. Based on the volumetric method, the Hirst-type sampler was used according to the European recommendations [23]. The station is located on the roof of the Collegium Śniadeckiego building, 20 meters above ground level and 200 meters above sea level (50° 3' 49" N; 19° 57' 19" E).

The pollen grains were sucked into a rotating drum covered with transparent tape (Melinex tape) with an adhesive fluid, which was changed once a week and then divided into seven segments corresponding to 24-hour periods. The tape fragments prepared in this way are placed on microscope slides, secured in a mixture of glycerin and gelatin with phenol (gelvatol) added, and then stained with basic fuchsin. The samples were examined using a light microscope at 400× magnification. Pollen grains were counted along 4 horizontal transects in Krakow. This method meets the requirement to count the minimum surface examined 10% of the entire deposition area [24]. The number of pollen counted in all horizontal lines is uploaded to the online database stored on the server of Jagiellonian University Medical College [25]. Pollen concentrations were automatically recalculated and expressed as pollen grains per cubic meter of air, per 24 hours (Pollen/m$^3$).

The daily birch and grasses pollen concentrations obtained during the pollen seasons defined as the periods when pollen is present in the air were used in the study. As the season beginning, we assessed the first day with pollen concentration above zero, while the last day with pollen was considered as the season end. This definition of the season is consistent with [26]. All days with zero pollen count within the pollen seasons were also included into the analyses.

Fig 1 shows the distribution of daily birch and grass pollen concentrations in the years 1991 - 2024. The distribution of pollen concentration during the pollen season is more heterogeneous for grasses, so the creation of separate predictive models for birch and grasses was required. In addition, the pollen season dynamics of both studied taxa differ significantly over the years, making the prediction more challenging.

Table 1 describes the basic statistics of the pollen concentration data. The different number of data points results from the fact that data outside the pollen seasons are omitted, and the duration of the pollen seasons is longer in the case of grasses.

The meteorological data fully cover the pollen concentration data in daily intervals. Data were collected from the European Climate Assessment and Dataset website [27] as blended data in ASCII file format. All observations were obtained from the Krakow-Balice station located 233 meters above sea level, and its coordinates are 50° 4' 49" and 19° 48' 6" E. The horizontal distance between the weather station and the pollen collection station is 11.11 kilometers and the height difference is 30 meters.

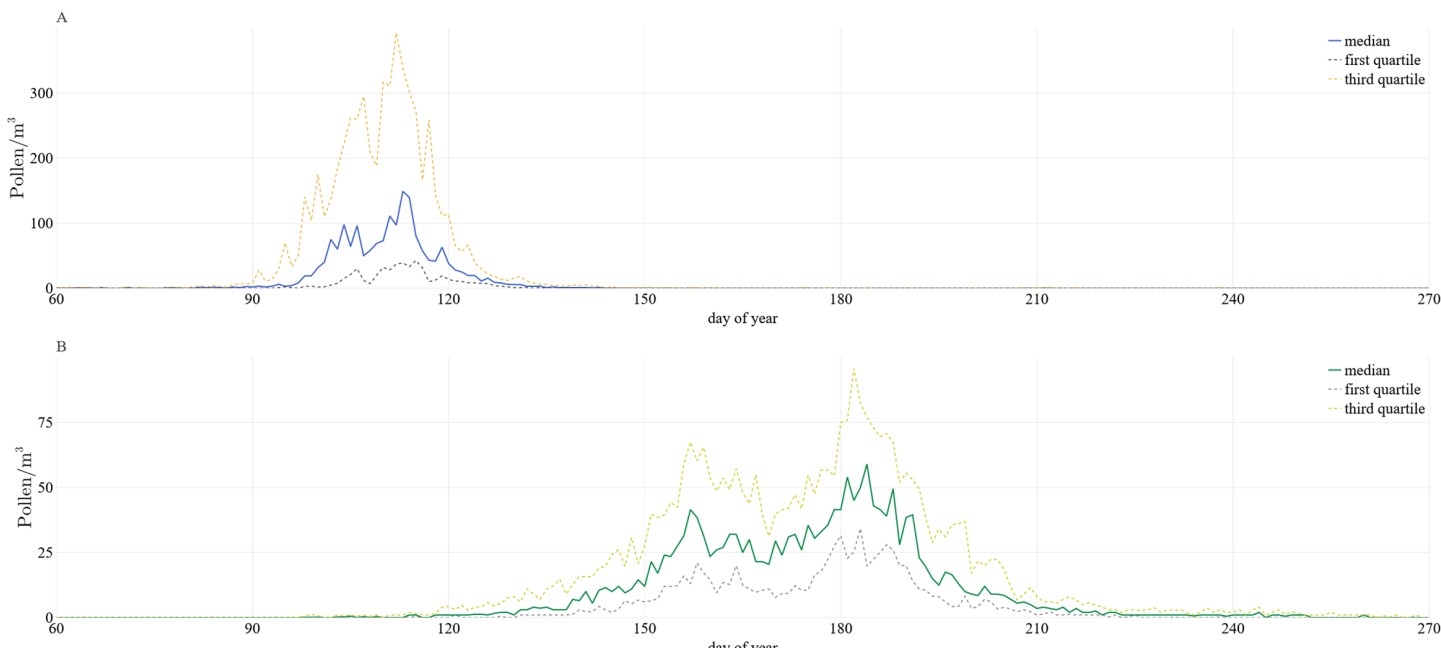

**Fig 1**. **Daily birch and grass pollen concentration.** Birch (A) and grasses (B) pollen concentration data collected at the Jagiellonian University Collegium Medicum in 1991 - 2024. The solid line shows the median pollen count for each day of the year, while the dashed lines denote the first (25th percentile) and third (75th percentile) quartiles, illustrating the typical seasonal peak and interannual variability. Periods outside the pollen seasons have been shortened to improve readability.

**Table 1**. **Descriptive statistics of the daily birch and grass pollen concentrations used into the study.**

| taxon | data points | feature | mean | std | min | median | max | distribution |
|-------|-------------|---------|------|-----|-----|--------|-----|--------------|
| *Betula* | 3425 | date | – | – | 1991-04-03 | 2003-08-15 | 2024-05-11 | uniform |
| | | Pollen/m$^3$ | 59.902 | 232.874 | 0 | 1 | 4199 | non-normal (p < 0.001) |
| Poaceae | 5395 | date | – | – | 1991-05-10 | 2008-05-05 | 2024-08-31 | uniform |
| | | Pollen/m$^3$ | 15.892 | 29.6273 | 0 | 3 | 437 | non-normal (p < 0.001) |

std - standard deviation, min - minimum value, max - maximum value.

To our knowledge, based on reports on the meteorological data in both stations (Balice and Kraków city center) the differences between them are relatively low. According to [28], in 2001-2010 annual temperature in Kraków center was higher by 0.7 °C in comparison to Balice station, whereby the differences were observed mainly in winter, while the pollen seasons are finished. The other data, like sunshine, cloud cover, annual relative humidity (77% vs 78%, Kraków and Balice, respectively), were slightly different.

We have selected 10 meteorological features that are statistically described in Table 2.

Shapiro-Wilk test from the HypothesisTests.jl package [29] was used to test the normality of the data distribution. It performs a test of the null hypothesis that the data come from a normal distribution. The Shapiro–Wilk test indicated that both the daily pollen concentration data and the meteorological data used in this study deviate significantly from a normal distribution.

To provide a minimal data set necessary to replicate our study, two csv files with *Betula* pollen and two files with Poaceae pollen data obtained in 2022 and 2023, and the meteorological data of ten selected factors were uploaded to the Open Science Framework with DOI number: DOI 10.17605/OSF.IO/9YZCF [30].

**Table 2**. Statistical description of meteorological data for birch and grass pollen seasons.

| taxon | feature | unit | mean | std | min | median | max | distribution |
|---|---|---|---|---|---|---|---|---|
| *Betula* | mean temperature | 0.1°C | 132.301 | 59.494 | -136.0 | 139.0 | 281.0 | non-normal (p < 0.001) |
| | minimum temperature | 0.1°C | 79.4 | 56.151 | -166.0 | 85.0 | 206.0 | non-normal (p < 0.001) |
| | maximum temperature | 0.1°C | 188.682 | 69.186 | -112.0 | 196.0 | 351.0 | non-normal (p < 0.001) |
| | humidity | 1% | 37.747 | 36.838 | 0.0 | 53.0 | 98.0 | non-normal (p < 0.001) |
| | cloud cover | okta | 4.269 | 2.154 | 0.0 | 5.0 | 8.0 | non-normal (p < 0.001) |
| | sunshine duration | 0.1 hour | 41.804 | 45.016 | 0.0 | 21.0 | 152.0 | non-normal (p < 0.001) |
| | mean wind speed | 0.1 m/s | 15.786 | 20.193 | 0.0 | 10.0 | 242.0 | non-normal (p < 0.001) |
| | mean sea level pressure | 0.1 hPa | 10061.021 | 984.711 | 47.0 | 10159.0 | 10412.0 | non-normal (p < 0.001) |
| | global radiation | $W/m^2$ | 152.026 | 81.385 | 15.0 | 125.0 | 341.0 | non-normal (p < 0.001) |
| | snow depth | 1 cm | 0.157 | 1.39 | 0.0 | 0.0 | 20.0 | non-normal (p < 0.001) |
| Poaceae | mean temperature | 0.1°C | 160.739 | 48.353 | -29.0 | 165.0 | 281.0 | non-normal (p < 0.001) |
| | minimum temperature | 0.1°C | 108.355 | 45.422 | -78.0 | 112.0 | 222.0 | non-normal (p < 0.001) |
| | maximum temperature | 0.1°C | 218.343 | 58.706 | 2.0 | 224.0 | 373.0 | non-normal (p < 0.001) |
| | humidity | 1% | 39.94 | 37.581 | 0.0 | 59.0 | 98.0 | non-normal (p < 0.001) |
| | cloud cover | okta | 4.091 | 2.147 | 0.0 | 4.0 | 8.0 | non-normal (p < 0.001) |
| | sunshine duration | 0.1 hour | 32.784 | 43.318 | 0.0 | 12.0 | 153.0 | non-normal (p < 0.001) |
| | mean wind speed | 0.1 m/s | 15.002 | 19.031 | 0.0 | 10.0 | 275.0 | non-normal (p < 0.001) |
| | mean sea level pressure | 0.1 hPa | 10063.074 | 982.175 | 56.0 | 10160.0 | 10368.0 | non-normal (p < 0.001) |
| | global radiation | $W/m^2$ | 134.308 | 77.854 | 15.0 | 102.0 | 343.0 | non-normal (p < 0.001) |
| | snow depth | 1 cm | 0.002 | 0.058 | 0.0 | 0.0 | 3.0 | non-normal (p < 0.001) |

std - standard deviation, min - minimum value, max - maximum value.

**Input data preparation.** Each data file was converted to the CSV format and pre-processed using the DataFrames.jl package [31].

We used two data pre-processing strategies depending on the machine learning method being used.

For models that can take time series data as input (for example, recurrent neural networks), we used the past pollen concentration data for an appropriate interval (1-20 days, depending on the experimental variant) preceding the predicted day by *n* days. For example, for time window 14, to predict 4 days in the future, we used data from the interval between 18 and 4 days in the past. We used the same approach for meteorological data, including the weather forecast simulation, so we included meteorological data for the predicted day and *n* days earlier. In the experiments, the weather forecast simulation used historical meteorological data rather than an external forecast model. Specifically, for time window *w* and prediction horizon *m*, we treated the recorded daily meteorological values on the target day and the preceding $m + w$ days as forecast inputs. This design isolates the performance of the pollen-forecasting algorithms from uncertainties in real-time weather predictions.

We used rolling windows and moving averages for models that prefer tabular data (for example, decision trees). For each feature, additional columns were prepared that contained exponential moving averages (EMAs) [32] and their derivatives. The exponential moving average is given by Eq (1) where *t* is the size of the window, $\alpha$ is a smoothing factor, and $x_t$ is a feature value at point *t*.

$$EMA_t = \frac{x_t + (1 - \alpha)x_{t-1} + (1 - \alpha)^2 x_{t-2} + ... + (1 - \alpha)^t x_0}{1 + (1 - \alpha) + (1 - \alpha)^2 + ... + (1 - \alpha)^t} \tag{1}$$

Let *i* be an index of a data point, and *f* be a feature name, then the following features were constructed:

1. $ema_3 = EMA(data[f][(i - 2) : i])$
2. $ema_7 = EMA(data[f][(i - 6) : i])$
3. $ema_{20} = EMA(data[f][(i - 19) : i])$

4. $ema_{1/3} = \frac{data[f][i]}{ema_3}$

5. $ema_{3/7} = \frac{ema_3}{ema_7}$

6. $ema_{7/20} = \frac{ema_7}{ema_{20}}$

Similarly, given a time window $w$, to predict $n$ days in the future, we used data from the interval between $n + w$ and $n$ days in the past. For meteorological data, we also included preprocessed meteorological data for the predicted day and $n$ days before.

**Target variable and measures.** In this work, the main goal of predictive models is to forecast pollen concentrations 1, 4, and 7 days into the future using supervised learning. Since the target variable is numeric, this is a regression problem. To assess the quality of the regression, the mean absolute error (MAE) described by Eq (2) and root mean square error (RMSE) described by Eq (3) metrics were used.

$$\text{MAE}(y, \hat{y}) = \sum_{i=1}^{N} |y_i - \hat{y}_i| \tag{2}$$

$$\text{RMSE}(y, \hat{y}) = \sqrt{\frac{\sum_{i=1}^{N}(y_i - \hat{y}_i)^2}{N}} \tag{3}$$

In these equations, $y$, $\hat{y}$, $\bar{y}$, and $N$ denote a reference target value, a predicted target value, the mean of the target variable, and the number of observations, respectively.

However, from a clinical point of view, it is more important to classify pollen concentration into a specific category, allowing appropriate preventive measures to be taken. Three selected categories of daily pollen concentrations that cause allergic rhinitis symptoms were modified according to personal observation of the symptoms in Krakow patients [14]. So, from a technical point of view, this is a classification problem in which the target variable is categorical, with three possible classes for each taxon. The categories, along with the frequency of occurrence for each class, are as follows:

1. *Betula*
   (a) low: $1 - 10$ Pollen/m$^3$, 76.0%,
   (b) medium: $11 - 75$ Pollen/m$^3$, 12.8%,
   (c) high: $>75$ Pollen/m$^3$, 11.2%,
2. Poaceae
   (a) low: $1 - 10$ Pollen/m$^3$, 68.3%,
   (b) medium: $11 - 50$ Pollen/m$^3$, 22.6%,
   (c) high: $>50$ Pollen/m$^3$, 9.1%.

To assess the quality of classification for each class, we used the accuracy defined by Eq (4) where $TP$, $TN$, $FP$, $FN$ denote a true positive value, a true negative value, a false positive value, and a false negative value, respectively.

$$accuracy = \frac{TP + TN}{TP + TN + FP + FN} \tag{4}$$

## Machine learning methods

Machine learning is used to model phenomena based only on data. It is particularly useful for problems in which the relationship between parameters is complicated or the functional relationship between input and output variables is unknown [21].

There are a huge number of machine learning methods that can make predictions and analyze pollen concentrations based on historical data. Popular choices are linear models, boosted trees, and deep neural networks [10,12–15]. We selected several popular and representative methods that proved their usefulness in the prediction of the pollen season. In addition, we include MAGN, which has proven utility in data mining and prediction problems [33]. Table 3 describes the selected models along with information about the origin of the source code. Except for linear regression, the machine learning methods used in this study are nonparametric and do not require parametric assumptions, making them more robust to violations such as nonnormality.

The models were used with default hyperparameter settings, and their values for individual models are available in the supplementary material (S2 File).

The following subsections briefly describe each of the models listed above.

**Multi-associative graph network.** Multi-Associative Graph Networks (MAGN) [33] is a novel graph-based approach to representing and processing large-scale training data along with key relationships between data elements. Unlike conventional feedforward neural networks, where backpropagation is the primary training mechanism, MAGN uses a recursive, feedback-oriented graph structure inspired by how the human brain stores and retrieves information. This design makes incorporating new data on the fly easier and adapting existing models without a complete retraining phase.

The features are represented as sensory fields composed of sensory neurons using a dedicated structure called ASA-graphs [48]. This data structure vertically relates feature values, aggregates duplicates, and is seamlessly combined with other MAGN neurons. The data neurons in the graph represent objects from the database, whereas the connections capture existing and newly discovered relationships. A tuning algorithm further refines the graph by learning which neurons (and their associated objects) should be prioritized, improving the network capacity for more accurate classification and more substantial relational dependencies. Overall, MAGN aims to offer a flexible, brain-inspired architecture that can quickly handle new information, preserve and exploit relational structures, and enable more efficient computational intelligence processes than standard feed-forward systems.

Fig 2 illustrates how MAGN encodes numerical and categorical information into a structured, multi-layer graph that supports learning through association rather than gradient-based optimization. Each sensory field (A and B) corresponds to a specific input feature, and the sensory neurons within them (A.1, A.2, A.3; B.1, B.2, B.3) represent distinct observed feature values aggregated across the dataset. The duplicate counter beneath each sensory neuron reflects the frequency of that value, allowing MAGN to prioritize more statistically relevant information. Object neurons (O.1, O.2, O.4) serve as integrative units connecting the individual feature values that co-occur within a training instance. Defining connections (solid arrows) establish the composition of an object in the feature space, whereas similarity connections (dotted lines) capture the statistical co-occurrence of feature values across many objects. This relational structure enables MAGN to

**Table 3**. **Overview of the machine learning methods evaluated in this study.** For each model, the table lists its methodological family, canonical theoretical reference, and the specific software implementation used in our experiments. The theoretical references point readers to foundational publications describing each method, while implementation references provide links to the exact packages and libraries used to ensure reproducibility.

| family | name | theory | implementation |
|---|---|---|---|
| lazy | K-Nearest Neighbors | [34] | NearestNeighborModels.jl [35] |
| | MAGN | [33] | witchnet [36] |
| linear | Linear Regression | [37] | MLJLinearModels.jl [38] |
| tree-based | Decision Trees | [39] | DecisionTree.jl [40] |
| | Random Forest | [41] | DecisionTree.jl [40] |
| | XGBoost | [42] | XGBoost.jl [43] |
| deep learning | Convolution | [44] | Flux.jl [45] |
| | LSTM | [46] | Flux.jl [45] |
| | GRU | [47] | Flux.jl [45] |

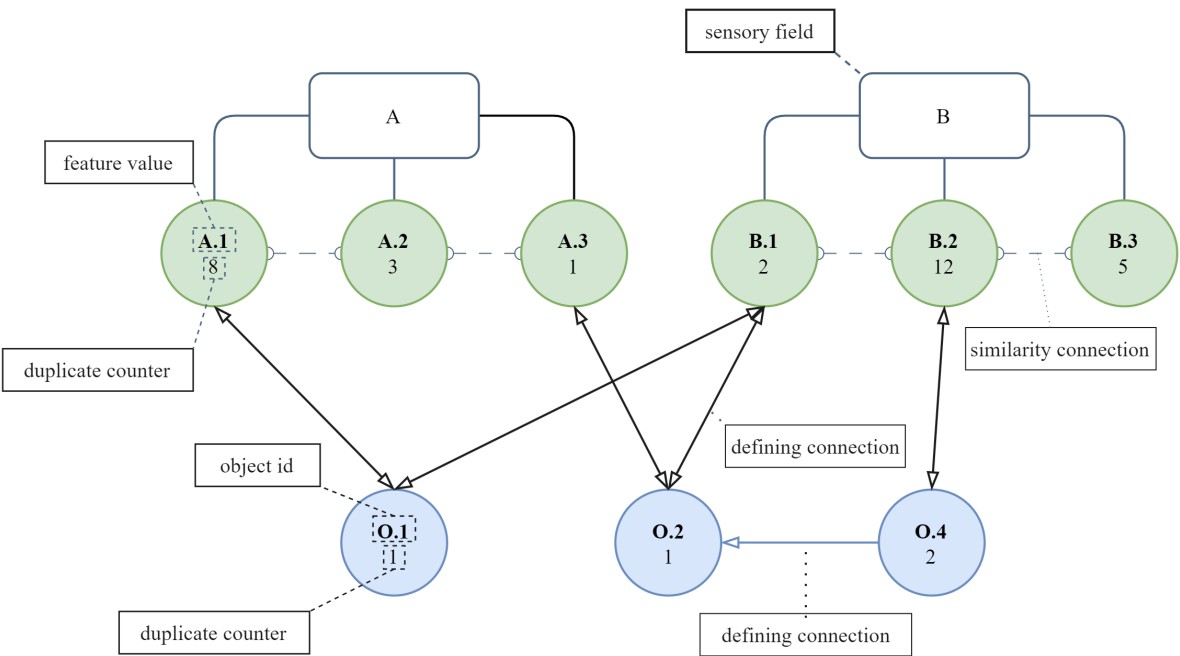

**Fig 2. MAGN structure.** Schematic structure of the Multi-Associative Graph Network (MAGN). Sensory fields (rectangular nodes A and B) contain sensory neurons representing unique feature values (green circles), each with an associated duplicate counter indicating how many times the value appeared in the training data. Object neurons (blue circles) represent individual data samples and are linked to the sensory neurons through defining connections, which encode the feature composition of each object. Similarity connections between sensory neurons capture statistical associations between feature values across the dataset, while duplicate counters on object neurons reflect the frequency of feature values and their patterns. Together, these components form a hierarchical associative graph that enables incremental learning, relational reasoning, and efficient retrieval.

retrieve patterns by following associative paths rather than performing iterative optimization, which explains its strong performance across accuracy, MAE, execution time, and memory metrics. The interactions between sensory neurons, object neurons, and associations enable MAGN to store training data, discover new relationships, and generalize over previously seen data.

This work used the lazy regressor and classifier based on neural activation propagation in MAGN for predictions. It utilized similarity connections to fuzzy inputs represented by the sensory neurons. With fuzzy activation, the signal propagates to data neurons by defining connections. Then, an algorithm called similarity voting estimates the target value using different techniques depending on the type of target variable (classification for categorical variables and regression for numerical variables).

In addition, due to its efficient structure, MAGN was used to calculate mutual information [49] to analyze the relationship between input and target features. Mutual information measures how much knowledge of one random variable reduces the uncertainty about another. Mathematically, mutual information is defined by Eq (5) [50]. Here, $P_{(X,Y)}$ is the joint probability distribution for the pair of random variables $X$ and $Y$, both defined in the same probability space. At the same time, $P_X$ and $P_Y$ are the marginal distributions for $X$ and $Y$, respectively.

$$MI(X, Y) = \sum_{x \in X} \sum_{y \in Y} P_{(X,Y)}(x, y) \log \frac{P_{(X,Y)}(x, y)}{P_X(x) P_Y(y)} \tag{5}$$

MAGN was also used to extract frequent patterns and association rules to characterize the pollen season. A frequent pattern refers to a combination of feature values (captured by the sensory neurons) that appears at least as often as the specified minimum support threshold [51]. A support of pattern $A$ measures how frequently a pattern appears in a dataset to be considered significant and is defined by Eq (6).

$$\text{support}(A) = \frac{counter(A)}{counter(objects)} \tag{6}$$

An association rule is an if-then statement of the form $A \rightarrow B$, where $A$ and $B$ are frequent patterns [52]. It indicates that whenever pattern $A$ appears in an object, pattern $B$ is also likely to appear. Frequent patterns and association rules are highly effective in data mining to uncover hidden relationships within datasets. For a given rule $A \rightarrow B$, the confidence measures the strength of the implication by calculating the conditional probability that pattern $B$ occurs given that pattern $A$ occurs [53]. Confidence is a measure of certainty of a rule and is defined as in Eq (7) where $P(B \mid A)$ is the conditional probability (the probability of an event occurring, given that another event is already known to have occurred).

$$\text{confidence}(A \rightarrow B) = P(B \mid A) \tag{7}$$

Lift measures the strength of association between the feature values in a rule compared to what would be expected if they were independent [54]. Lift is defined by Eq (8).

$$\text{lift}(A \rightarrow B) = \frac{P(B \mid A)}{P(B)} \tag{8}$$

**K-Nearest neighbors.** The k-Nearest Neighbors (k-NN) algorithm [34] is a non-parametric classification and regression method that assigns target values to predicted data based on the values of the target variable of the k-closest training samples in the n-dimensional feature space. It calculates the distance (commonly using the Euclidean distance) between the query point and all training points to determine the nearest neighbors. k-NN is a lazy algorithm, meaning that it does not build a model but instead makes predictions at runtime based on the raw training data. The choice of k is crucial: a small k may lead to overfitting, while a large k may oversmooth the decision boundaries [55]. Despite its simplicity, k-NN performs well in low-dimensional spaces but struggles with high-dimensional data due to the curse of dimensionality [56] while all dimensions influence results in the same way, even if some of them are irrelevant. There is no attention mechanism that could prioritize more essential data features.

The k-NN algorithm has also been widely analyzed in the context of practical data pre-processing requirements. In many applications, the performance of k-NN depends strongly on feature scaling, because distance metrics such as Euclidean or Manhattan distance are sensitive to differences in feature ranges. Standardization or normalization is therefore typically applied to ensure that no single variable disproportionately influences the computed distances [57]. Additionally, various distance metrics can be used to better adapt k-NN to specific data distributions or to reduce the impact of correlated features. These enhancements enable the algorithm to be more robust in heterogeneous datasets, such as meteorological time series combined with biological variables.

**Linear regression.** Linear regression is a fundamental machine learning method used to model the relationship between a dependent variable $y$ and $n$ independent variables $x_n$ by fitting a linear equation Eq (9) [37]. The parameters ($\beta$ coefficients) are usually estimated by minimizing the sum of squared residuals (residual is the difference between the predicted value and observed value, e.g., $y_i - \hat{y}_i$). Despite its simplicity and interpretability, linear regression assumes linearity, independence, homoscedasticity (an assumption of equal or similar variances in different groups), and normality of

residuals, making it sensitive to outliers and multicollinearity [58].

$$y = \beta_0 + \beta_1 x_1 + \beta_2 x_2 + ... + \beta_n x_n \tag{9}$$

In practical applications, several extensions of classical linear regression are used to mitigate its limitations and improve predictive performance. Techniques such as regularization, including Ridge (L2) and Lasso (L1) regression [59], add penalty terms to the loss function to reduce overfitting and address multicollinearity by shrinking or eliminating coefficients. These modifications help stabilize parameter estimates and improve generalization, particularly when predictors are highly correlated or when the number of features is large relative to the number of observations. Regularized linear models have been widely used in environmental and atmospheric sciences due to their robustness and ability to handle noisy real-world data [15].

**Tree-based methods.** A decision tree is a supervised machine learning algorithm structured as a tree where each internal node represents a decision based on a feature, each branch represents a subset of feature values, and each leaf node represents a predicted class or value [39]. The tree is constructed using algorithms such as ID3 or CART, which recursively split data based on criteria such as information gain, Gini impurity, or variance reduction [60]. Decision trees are interpretable and can handle both numerical and categorical data. However, they are prone to overfitting, especially with deep trees, which can be mitigated using pruning or more advanced models such as gradient boosting and random forests.

Random Forest is an ensemble learning method that builds multiple decision trees and combines their results to improve predictive accuracy and reduce overfitting [41]. It works by training each tree on a random subset of the data and selecting a random subset of features at each split to enhance diversity. Predictions are made by voting among trees. The model is highly robust to overfitting and works well with large datasets, but it is less interpretable than simpler models. Random Forest is widely used in various applications due to its versatility, scalability, and ability to effectively handle missing data and imbalanced datasets [61].

XGBoost (Extreme Gradient Boosting) [42] is a machine learning model based on gradient boosting. This technique assumes that the next model minimizes the overall prediction error when combined with previous models. It builds an ensemble of decision trees sequentially, where each new tree corrects the errors of the previous ones using gradient-based optimization. XGBoost incorporates regularization techniques (such as L1 and L2 penalties) to prevent overfitting, making it more robust than traditional gradient boosting methods. Due to its high predictive accuracy and execution speed, XGBoost has become a dominant choice in machine learning competitions and real-world applications, including medical diagnosis [62].

**Deep neural networks.** Deep neural networks (DNNs) are multilayer artificial neural networks that consist of multiple hidden layers between the input and output layers. Using techniques such as backpropagation and activation functions, DNNs have achieved state-of-the-art performance in various fields, including time series prediction [44].

Convolution is a mathematical operation used in signal processing and deep learning, where a filter (kernel) slides over input data to extract features by computing element-wise multiplications and summing the results [44]. In deep learning, 1-dimensional convolutional layers help capture temporal dependencies in sequences. The convolution operation is efficient because it enables weight sharing and local connectivity, significantly reducing the number of parameters compared to fully connected layers [63].

Long Short-Term Memory (LSTM) is a type of recurrent neural network (RNN) designed to overcome the vanishing gradient problem, allowing it to learn long-term dependencies in sequential data [46]. LSTMs use input, forget, and output gates to regulate the flow of information, ensuring that relevant data is retained while irrelevant information is discarded. The cell state acts as a memory unit, allowing LSTMs to store and update information across long sequences, making them well-suited for time series forecasting. Despite their effectiveness, LSTMs can be computationally expensive, which reduces their practical application.

Similarly to LSTMs, Gated Recurrent Units (GRUs) are a type of RNN designed to capture long-term dependencies in sequential data while addressing the problem of vanishing gradients [47]. Unlike LSTMs, GRUs use only two gates: the reset and update gates, making them computationally more efficient. The update gate determines how much past information should be carried forward, while the reset gate controls how much new information should be incorporated. GRUs have been widely applied in time series forecasting, often performing comparably to LSTMs with fewer parameters [64], effectively reducing computational complexity without performance sacrifice.

Fig 3 shows the architecture of the deep neural network used in the experiments.

## Experiments

Two experiments were carried out independently for birch and grass pollen:

1. Experiment 1: prediction of predefined pollen concentration classes (low, medium, high) 1, 4, and 7 days ahead,
2. Experiment 2: characterization of pollen seasons using machine learning methods.

The experiments were carried out using the MLJ.jl package [65]. The following subsections describe each experiment in detail.

**Experiment 1.** The goal of Experiment 1 was to systematically compare the forecasting accuracy of nine machine learning algorithms listed in Table 3 across three forecast horizons (1, 4, and 7 days ahead).

For each forecast feature vectors were prepared as detailed in the Input Data Preparation section.

We evaluated each model on two parallel tasks:

- **Regression**: predicting the continuous pollen concentration, assessed via Root Mean Squared Error (RMSE) and Mean Absolute Error (MAE).
- **Classification**: predicting discrete pollen concentration categories, assessed by overall accuracy as described in the Target Variable and Measures subsection.

To prevent temporal leakage, we grouped records by calendar year and randomly assigned 70% of years to the training set and the remaining 30% to the independent test set, ensuring no overlap in years between partitions.

All algorithms were trained using their default hyperparameter settings to maintain fair comparability. Performance metrics were computed exclusively on the held-out test set for each horizon.

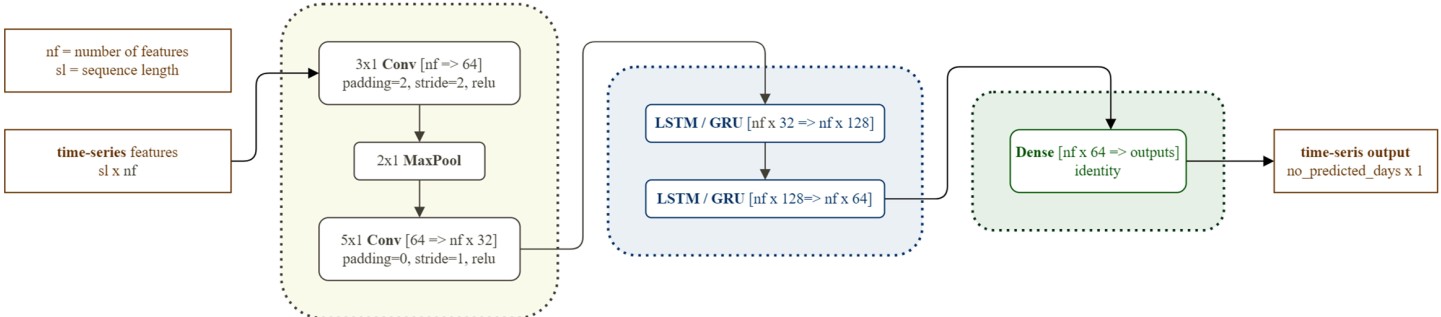

**Fig 3. Deep neural network architecture.** The input features, such as pollen concentration or meteorological data, are in the form of time series. The convolutional layers, followed by the LSTM/GRU layers, and the dense layers were used sequentially. The output is a time series predicting the pollen concentration in the following days.

This design allowed us to identify which methods most effectively capture both the numeric trends and categorical thresholds of birch and grass pollen seasons under varying lead times.

**Experiment 2.** Building on the predictive performance established in Experiment 1, Experiment 2 aimed to interpret and characterize the relationships between the input and target variables. It comprised three components:

- **Feature importance analysis**: feature importance learned by the top-performing model to rank each predictor's contribution to model accuracy.
- **Mutual information assessment**: normalized mutual information between each input variable and the target (pollen concentration or category) to quantify their statistical dependence without assuming linear relationships.
- **Association rule mining**: association rules mined by MAGN uncover patterns of meteorological and pollen variables that precede high or low pollen days.

These analyses collectively reveal which factors and combinations of them most strongly influence pollen forecasts and offer an interpretable characterization of pollen season dynamics.

**Software and hardware.** All experiments were implemented in the Julia programming language v1.11.3 [66]. The packages used are listed in the appropriate subsections above. The source code is publicly available under the MIT license [67].

The experiments were carried out on a dedicated workstation with one AMD Ryzen Threadripper Pro 5965WX CPU, 128 GB RAM, and 2 x GPU NVidia GeForce 3090. To ensure stable thermal conditions during testing, the CPU and GPUs are cooled with a custom water cooling system, and the workstation is placed in a server cabinet with a dedicated connection for mechanical ventilation.

## Results

An overview of the study design and main findings is shown in the Supporting information (S1 Fig).

### Experiment 1

Table 4 presents the results of the experiment separately for a given taxon and the number of predicted days ahead. The results are sorted by accuracy, but the winning model for each metric is marked in bold. For birch pollen, XGBoost and MAGN deliver the highest classification accuracy. For grass pollen, XGBoost, Random Forest, and MAGN lead in 1- and 4-day forecasts. However, for 7-day predictions, the Conv-LSTM DNN outperforms the others, likely reflecting its superior ability to capture complex, multidimensional temporal patterns.

Fig 4 shows the comparison of models based on the average results for accuracy, MAE, execution time, and total memory used during training and prediction. XGBoost and MAGN produced similarly superior performance compared to the other models overall, while MAGN proved the most efficient in terms of memory usage and runtime. This is achieved by the algorithm-as-a-structure approach, where multiple relations are represented in the structure of the neural network, eliminating the need to compute these relations in the training phase.

Fig 5 shows a graph of the observed and predicted grass and pollen concentrations predicted by the XGBoost model for the sample seasons. This model was selected because it had the best prediction accuracy for this sample period.

### Experiment 2

Fig 6 shows feature importance and normalized mutual information for all input variables in predicting pollen concentration. Across both birch and grass, past pollen levels are by far the strongest predictor, with temperature ranking second (relative normalized mutual information = 36.1%–65.0%).

A deeper look at relative normalized mutual information for seven lower-ranked meteorological drivers reveals that humidity contributes 7.7%–8.1%, sunshine duration 6.4%–9.5%, radiation 4.7%–6.4%, cloud cover 4.3%–5.3%, wind

**Table 4**. Pollen concentration forecasting results.

| taxon | days ahead | model | accuracy | mae | rmse | time | memory |
|---|---|---|---|---|---|---|---|
| *Betula* | 1 | **XGBoost** | **0.922** | 27.15 | 136.764 | 17.336 | 7340 |
| | | **MAGN** | 0.916 | 26.937 | 130.66 | **7.332** | **837** |
| | | **Decision Trees** | 0.894 | **26.625** | 130.846 | 13.999 | 7336 |
| | | K-Nearest Neighbors | 0.867 | 26.874 | 126.844 | 10.595 | 7338 |
| | | **Random Forest** | 0.841 | 28.494 | **124.383** | 9.314 | 7536 |
| | | Conv-LSTM DNN | 0.84 | 33.279 | 158.074 | 1602.9 | 214337 |
| | | Conv-GRU DNN | 0.719 | 37.767 | 159.795 | 3148.41 | 228738 |
| | | Linear Regression | 0.602 | 71.349 | 173.272 | 11.58 | 7337 |
| | 4 | **MAGN** | **0.883** | 39.327 | 160.51 | **4.342** | **561** |
| | | XGBoost | 0.879 | 40.024 | 176.681 | 12.709 | 7324 |
| | | Decision Trees | 0.864 | 47.092 | 207.021 | 8.078 | 7321 |
| | | **Conv-LSTM DNN** | 0.844 | **31.051** | 152.126 | 1602.9 | 2995 |
| | | K-Nearest Neighbors | 0.836 | 42.101 | 178.778 | 9.867 | 7435 |
| | | Conv-GRU DNN | 0.814 | 36.706 | 159.676 | 3148.41 | 228738 |
| | | **Random Forest** | 0.715 | 42.368 | **151.122** | 9.950 | 7477 |
| | | Linear Regression | 0.516 | 78.464 | 180.295 | 9.49 | 7325 |
| | 7 | **MAGN** | **0.872** | 42.069 | 168.523 | **2.02** | **484** |
| | | **XGBoost** | **0.872** | 51.319 | 208.498 | 12.586 | 7304 |
| | | Decision Trees | 0.862 | 64.129 | 257.572 | 11.525 | 7301 |
| | | K-Nearest Neighbors | 0.844 | 54.072 | 197.97 | 12.024 | 7377 |
| | | **Conv-LSTM DNN** | 0.83 | 40.005 | **159.748** | 2183.45 | 208284 |
| | | **Conv-GRU DNN** | 0.808 | **39.95** | 165.033 | 3148.41 | 228738 |
| | | Random Forest | 0.7 | 50.885 | 167.056 | 11.605 | 7398 |
| | | Linear Regression | 0.503 | 79.971 | 182.744 | 8.773 | 7306 |
| Poaceae | 1 | **Random Forest** | **0.861** | **7.864** | **20.464** | 20.245 | 10579 |
| | | **MAGN** | 0.857 | 8.852 | 25.32 | 14.549 | **737** |
| | | XGBoost | 0.848 | 8.109 | 21.109 | 25.60 | 10239 |
| | | **Decision Trees** | 0.827 | 9.313 | 22.651 | **12.987** | 10234 |
| | | Linear Regression | 0.805 | 11.373 | 23.236 | 23.80 | 10237 |
| | | Conv-LSTM DNN | 0.787 | 12.325 | 31.441 | 4509.94 | 390109 |
| | | K-Nearest Neighbors | 0.77 | 11.071 | 24.437 | 19.86 | 10237 |
| | | Conv-GRU DNN | 0.725 | 12.979 | 29.396 | 5493.77 | 422627 |
| | 4 | **XGBoost** | **0.818** | 9.895 | 25.43 | 22.58 | 10216 |
| | | **Random Forest** | 0.802 | 9.972 | **24.43** | 15.1031 | 10482 |
| | | **MAGN** | 0.802 | 10.598 | 27.36 | **7.851** | **665** |
| | | K-Nearest Neighbors | 0.799 | 10.725 | 26.53 | 19.424 | 10214 |
| | | Conv-LSTM DNN | 0.796 | 10.525 | 27.265 | 2995.14 | 391027 |
| | | Decision Trees | 0.786 | 11.318 | 27.232 | 20.315 | 10211 |
| | | Conv-GRU DNN | 0.762 | 11.83 | 27.84 | 5493.77 | 422627 |
| | | Linear Regression | 0.73 | 12.606 | 25.807 | 14.131 | 10213 |
| | 7 | **Conv-LSTM DNN** | **0.8** | **10.416** | 27.833 | 4509.94 | 390109 |
| | | **Random Forest** | 0.775 | 11.497 | **26.905** | 20.177 | 10351 |
| | | **MAGN** | 0.771 | 12.235 | 29.771 | **3.985** | **419** |
| | | K-Nearest Neighbors | 0.768 | 12.072 | 28.719 | 17.976 | 10183 |
| | | XGBoost | 0.766 | 11.876 | 28.316 | 18.549 | 10184 |
| | | Conv-GRU DNN | 0.754 | 12.268 | 29.294 | 5493.77 | 422627 |
| | | Decision Trees | 0.747 | 13.131 | 30.837 | 13.0 | 10182 |
| | | Linear Regression | 0.666 | 14.537 | 28.169 | 21.908 | 10182 |

time unit: second, memory unit: megabyte.

speed 2.0%–3.9%, sea level pressure 2.2%–3.0%, and snow depth 0.5%–1.6%. Although these secondary factors fall below past pollen and temperature in overall importance, they still make non-negligible contributions. This demonstrates

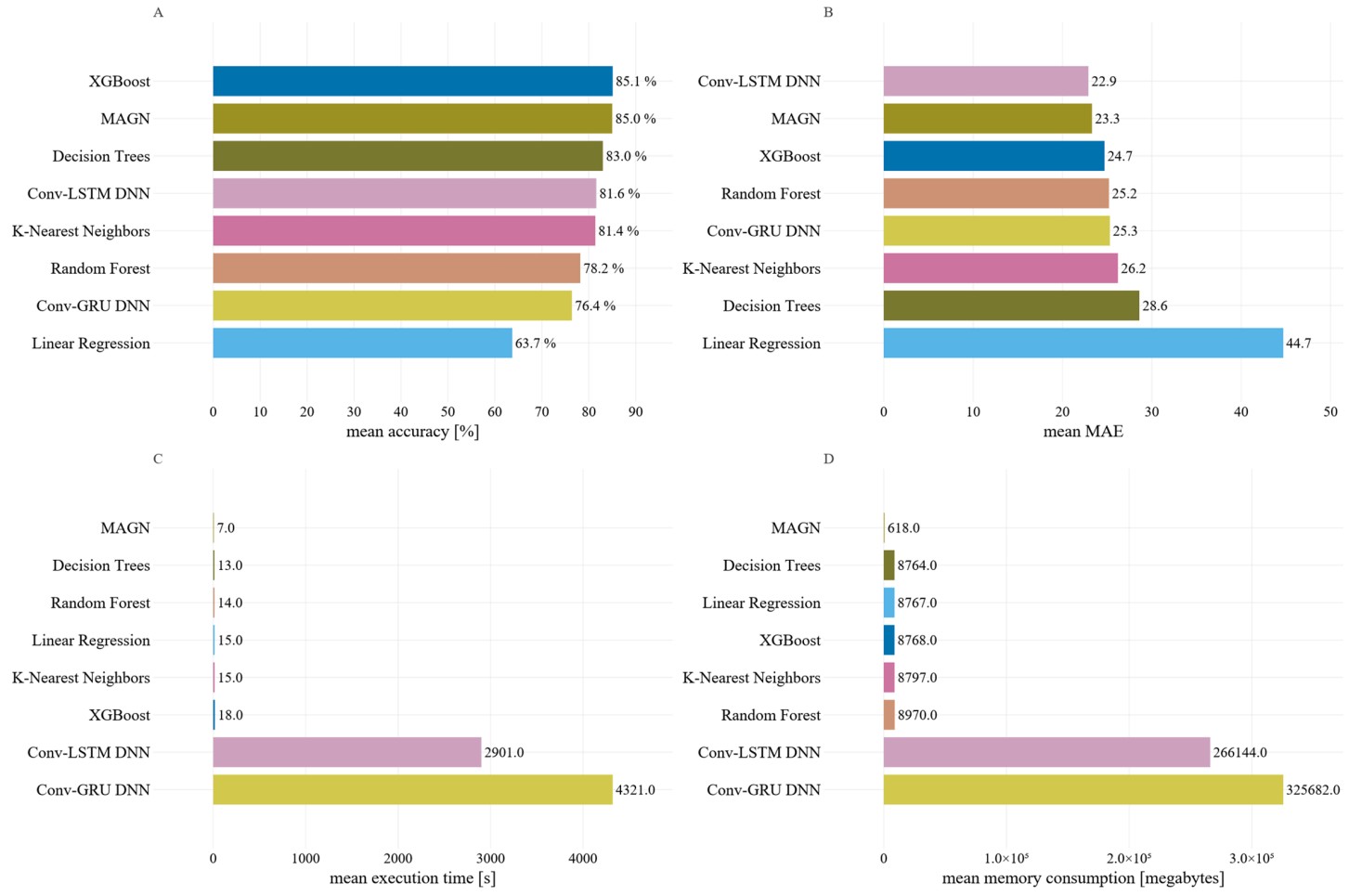

**Fig 4. Comparison of machine learning models based on metric averages.** The machine learning methods are compared using the averages of all variants of Experiment 1 for accuracy (A), MAE (B), execution time in seconds (C), and total memory used during training and prediction in megabytes.

that including a broader suite of environmental variables can further refine model performance under varying weather conditions.

Fig 7 illustrates the association rules linking combinations of meteorological variables from the previous three days with the resulting pollen concentration classes for *Betula* (A) and Poaceae (B). Several notable patterns emerge from the plot. First, the large circles located in the upper-right area of each panel represent rules with simultaneously high support and high confidence, indicating that certain combinations of temperature, humidity, cloud cover, wind speed, and sunshine duration frequently precede specific pollen concentration levels and do so with high reliability. The color scale further shows that many of these high-confidence rules also exhibit elevated lift values, meaning that the occurrence of these meteorological conditions increases the likelihood of the pollen class more than expected by chance.

A key point of interest is the presence of a few large bubbles with high confidence and support in both pollen types, suggesting that some meteorological patterns are strongly and consistently associated with typical seasonal peaks. For *Betula* (panel A), the highest-lift rules tend to cluster at moderate support values, implying that particularly strong meteorological triggers occur less frequently but are highly predictive when they appear. In contrast, Poaceae (panel B) shows

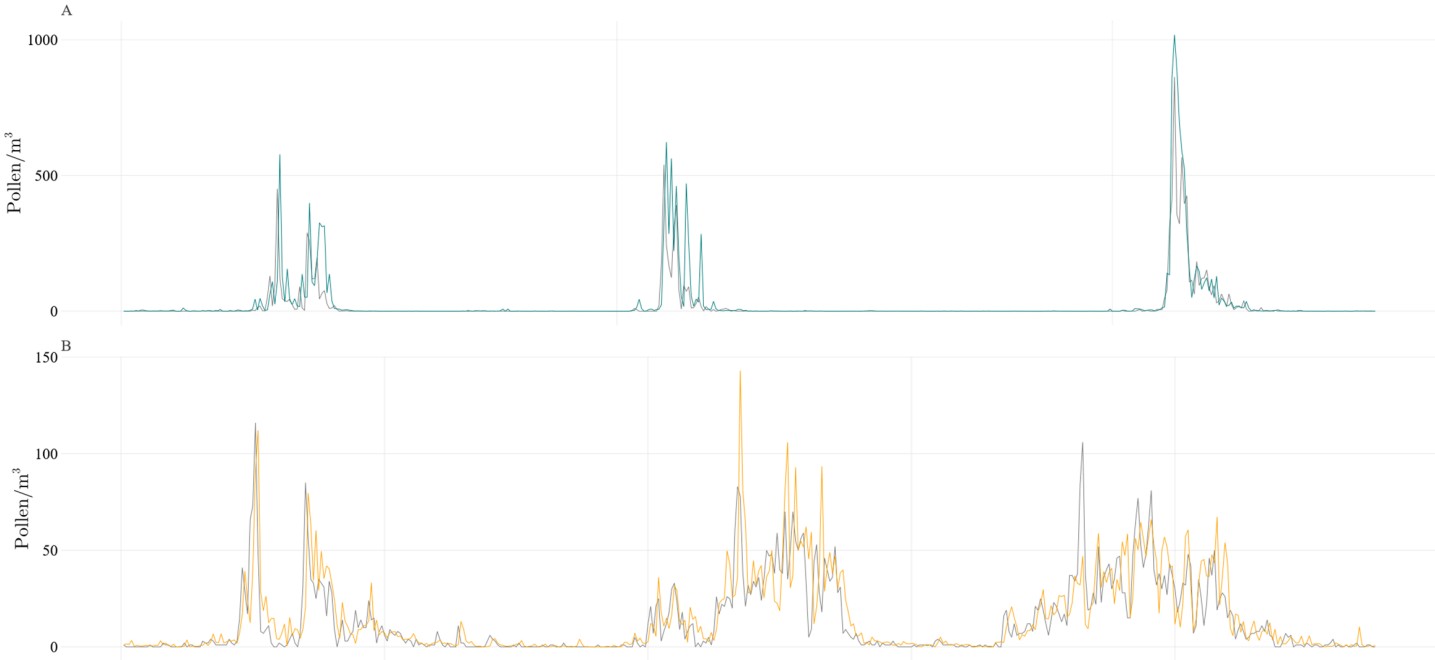

**Fig 5**. **Prediction examples.** The observed (gray) and predicted (green or orange) *Betula* (A) and Poaceae (B) pollen concentrations. The predictions were made by the XGBoost model for 3 randomly selected pollen seasons from the test set.

a broader spread of medium-to-high support rules, indicating that grass pollen is influenced by a wider variety of meteorological combinations. Together, these patterns demonstrate that association rule mining can effectively uncover both common and rare but highly predictive meteorological configurations, providing insights into how weather conditions shape pollen dynamics.

## Discussion

Predicting pollen concentrations, characterizing, and modeling upcoming pollen seasons are among the most critical goals of aerobiology [15] and are helpful for allergologists in clinical practice. Furthermore, tracking pollen concentrations is essential in clinical trials of specific immunotherapy [68]. The European Medicines Agency guideline on the clinical development of allergen immunotherapy products states that, for seasonal allergies, trials must record exposure to relevant allergens and specify in the study protocol the minimum pollen count necessary to establish both the baseline and evaluation periods [69]. Until now, machine learning methods have been used to predict changes in daily pollen concentrations [14,18,19,21], the severity of pollen-induced symptoms [70], daily alarm threshold [70]. As rightly emphasized in [15], studies in the analyzed topic differ significantly in terms of modeling techniques, predictor variables, and validation methods.

Building on these advances, our study is the first to compare seven widely used machine learning algorithms (K-Nearest Neighbors, Linear Regression, Decision Trees, Random Forest, XGBoost, Conv-LSTM DNN, Conv-GRU DNN) and Multi-Associative Graph Networks for retrospective daily birch and grass pollen forecasting over 34 years in Krakow. Unlike many studies, this work did not use temporal predictors such as day of the year, month of the year, and season of the year. However, a relatively large number of meteorological variables (ten) were used.

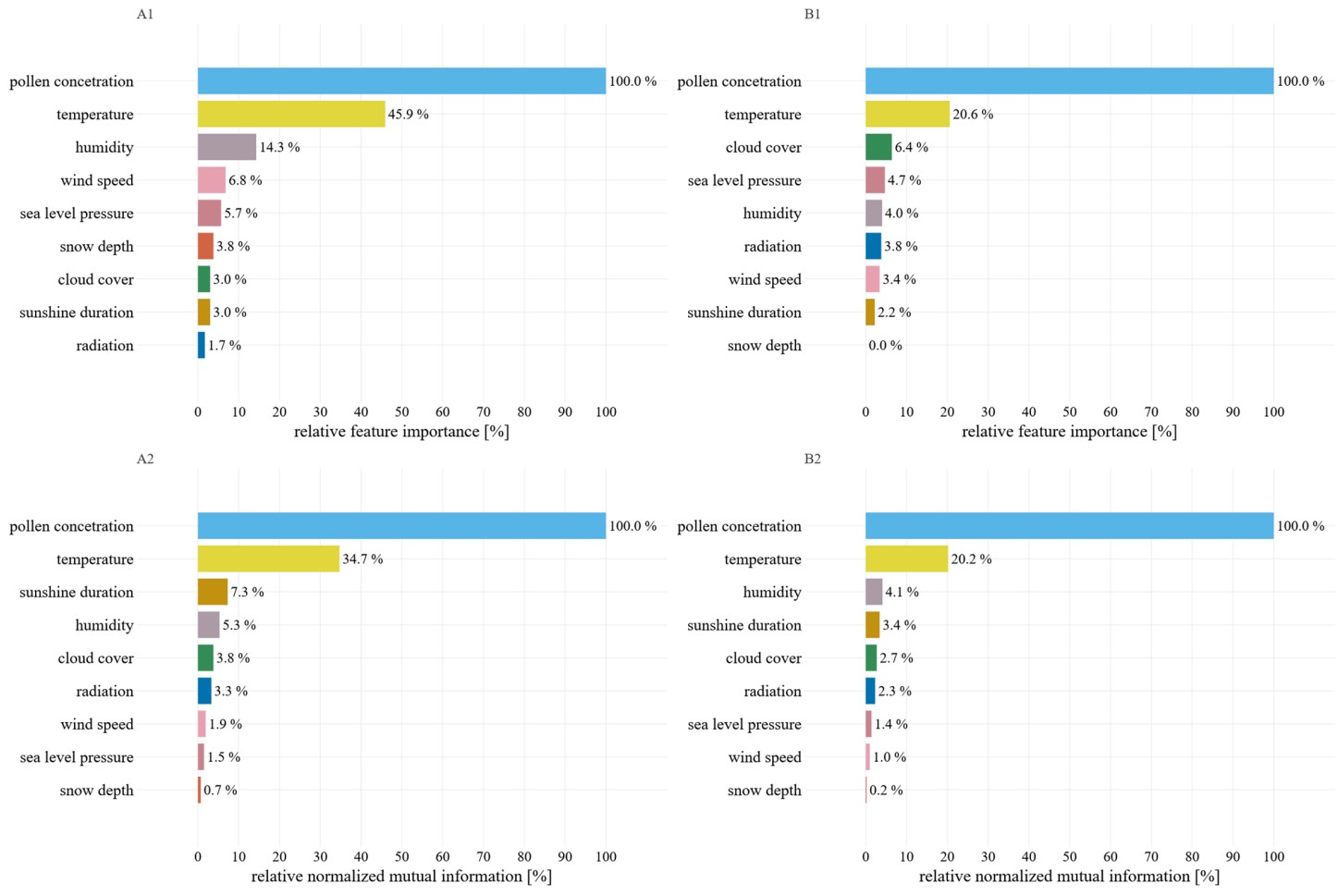

**Fig 6. Feature importance and normalized mutual information.** The top row shows the importance of the feature from the XGBoost model for *Betula* (A1) and Poaceae (B1). The bottom row shows the normalized mutual information for *Betula* (A2) and Poaceae (B2).

Our work's innovative nature is demonstrated by using a large original database containing a long pollen data series from 1991 to 2024 and by comparing the usefulness of different machine-learning methods in predicting and characterizing pollen seasons. The work was created as a collaboration between an interdisciplinary team consisting of an aerobiologist, machine learning experts, and practicing allergists. We compared the usefulness of many machine learning methods from various orthogonal families. To our knowledge, this is the only work that examines the use of associative knowledge graphs (MAGNs) to predict pollen concentration and the mining association rules that characterize pollen seasons [71].

The proposed experiments have shown that the best average efficiency in pollen concentration classification is achieved by XGBoost, followed by MAGN. These are effective and fast classifiers with a low memory footprint but sensitive to proper feature engineering. In turn, the best representation of the pollen concentration curve (regression task) was achieved by a deep neural network with convolutional layers and LSTM cells. These networks can effectively model time series, but their training and maintenance costs are high. The most time- and memory-efficient classifier was the one based on MAGN.

In order to guide future aerobiological modeling efforts, we have synthesized the practical trade-offs among the nine algorithms evaluated in this study. Table 5 provides a concise comparison of each method's principal advantages, such

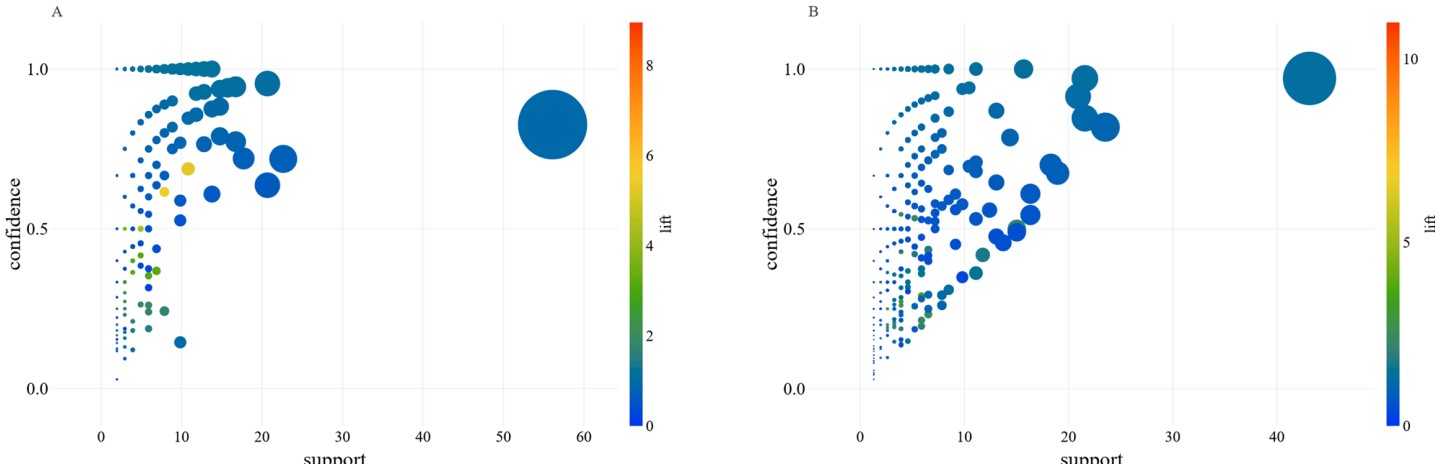

**Fig 7**. **Association rules analysis for meteorological data.** The association rules where the antecedent is the combination of average temperature, cloud cover, humidity, mean wind speed, and duration of the sunshine of the previous 3 days and the consequence is the pollen concentration class for *Betula* (A) and Poaceae (B). The bigger the dot, the bigger the support.

**Table 5**. **Comparative Overview of Machine Learning Methods for Pollen Forecasting.**

| Model | Advantages | Disadvantages |
|---|---|---|
| **Linear Regression** | •Highly interpretable<br>•Very fast training and inference | •Assumes linearity and normality<br>•Low accuracy |
| **K-Nearest Neighbors** | •No training phase<br><br>•Captures local non-linear patterns | •Prediction slow for large datasets<br><br>•Moderate accuracy |
| **Decision Trees** | •Interpretable via tree splits<br>•Fast inference | •Prone to overfitting<br>•Sensitive to data changes |
| **Random Forest** | •Good out-of-the-box performance<br><br>•Easy variable importance estimation | •Higher memory and compute than a single tree<br>•Less interpretable than a single tree |
| **XGBoost** | •State-of-the-art accuracy on pollen prediction<br>•Built-in regularization | •Complex hyperparameter tuning<br><br>•Higher computational cost than Random Forest |
| **Conv-LSTM DNN** | •Models complex, multidimensional temporal patterns<br>•End-to-end feature extraction | •Long training times<br><br>•High memory and hardware requirements |
| **Conv-GRU DNN** | •Similar benefits to Conv-LSTM with fewer parameters<br>•Faster convergence | •Still long training times<br><br>•Still high memory and hardware requirements |
| **MAGN** | •Comparable accuracy to XGBoost<br>•Fastest runtime and lowest memory footprint | •Novel, fewer off-the-shelf tools<br>•Requires graph construction step |

as predictive accuracy, interpretability, and resource efficiency-and their corresponding limitations, including distributional assumptions, computational cost, and ease of implementation. This overview enables researchers to select the most appropriate algorithm based on specific project requirements, data availability, and hardware constraints.

In our study, we used default hyperparameter settings for all machine learning models as the number of possible combinations of them surpasses our computational possibilities and the default settings were already optimized as usually

well-options. This choice was made to ensure fairness and comparability across a wide range of models and configurations. While more extensive hyperparameter tuning could potentially lead to improved performance for some methods, the high accuracy achieved with default settings demonstrates the robustness of the proposed framework. Future work could incorporate hyperparameter optimization techniques, such as grid search, evolutionary search, or Bayesian optimization, to further enhance performance.

In our work, we analyzed the use of different machine learning methods in assessing the impact of weather factors on the concentration of birch and grass pollen. The most important environmental variable that determines daily total pollen concentrations is the past daily total pollen concentration, which has been proven in our and other authors' work [10,15]. The literature emphasizes the influence of the mean daily temperature on a current or preceding day [72–74], maximum and minimum daily temperature [75,76]. These findings depend on the taxon studied, for example, in the case of birches, previous studies performed in Krakow confirmed that mean and minimum temperature and relative humidity explained the variability in daily pollen concentration of birch more effectively [13,14], while the dominant independent variable included in the models that predicted daily pollen concentration of grass was temperature [14]. Local microclimate conditions can have a strong influence on pollen release [77,78]. Therefore, the best solution is to develop forecasting models on a not-wide spatial scale, and only after a thorough evaluation of their effectiveness may they prove to be more universal [79]. In Switzerland, grass pollen concentration exhibits a positive relationship with temperature, as pollen levels tend to increase when the temperature exceeds 10° C. At the same time, both precipitation and humidity have a negative relationship with grass pollen, leading to decreased predicted pollen concentration with higher rainfall and humidity levels exceeding 70%.

In addition, analysis of association rules showed that there are many rules with high confidence, strong support, and significant support. This suggests that these combinations of parameters can effectively help predict pollen concentrations.

A key limitation of this study is the reliance on data collected solely from the Krakow region over a span of 34 years. While the long-term nature of the dataset strengthens temporal analysis, the geographic constraint may limit the direct applicability of the results to other regions with differing climatic and environmental conditions. However, the machine learning framework developed in this study is flexible and can be applied to other datasets, provided that region-specific pollen and meteorological data are available.

In the context of the limitations of the presented data pre-processing, it is also worth noting that the sliding-window approach inherently produces overlapping feature intervals across adjacent forecast dates, leading to high autocorrelation among lagged predictors. While nonparametric methods such as tree-based models and neural networks are relatively robust to multicollinearity, this overlap may still inflate apparent feature importance for highly autocorrelated variables. Future work could explore orthogonalization techniques or feature selection methods that mitigate such bias.

The practical implication of predicting pollen concentration is the possibility of intensifying antiallergic treatment and avoid activities outdoors at critical moments of pollen concentration to prevent the exacerbation of clinical symptoms in patients. An additional benefit of monitoring the pollen season is the precise assessment of the effectiveness of specific immunotherapy (AIT). The guidelines of the European Academy of Allergy and Clinical Immunology emphasize that monitoring daily pollen concentrations is necessary to determine the appropriate time window for assessing the effectiveness of AIT for seasonal allergic rhinoconjunctivitis (ARC) [68]. The first stage of our work was the construction and evaluation of models, which is presented in this publication. In the further stages of the project, we plan to use the presented models to build a mobile application that will predict pollen concentrations and will be made available to patients.

## Conclusion

Predicting and characterizing pollen seasons is a crucial aspect of aerobiology, with significant implications for allergy sufferers and public health. Our study contributes to this field by using a comprehensive historical database (1991–2024) and systematically comparing multiple machine learning methods for pollen concentration prediction. Our results confirm

that past pollen concentration is the single most powerful predictor. Among the meteorological predictors used, temperature emerges as the leading meteorological driver across taxa, whereas the remaining predictors (humidity, cloud cover, sunshine duration, mean wind speed, mean pressure at sea level, global radiation and snow depth) showed variable and significantly less significance. The application of associative knowledge graphs (MAGNs) and association rules for the characterization of the pollen season is an approach that has not been explored previously in this context.

Our study's principal methodological advance is the comprehensive evaluation of diverse machine-learning approaches, including the first application of MAGNs to pollen forecasting. Consistent with previous research, tree-based models (especially XGBoost) derived the highest predictive accuracy. Crucially, MAGNs achieved comparable performance while requiring the least memory and exhibiting the fastest runtimes among all models tested, making them particularly well suited for general deployment as well as for use on edge devices or in hardware-constrained settings. Conversely, deep neural architectures (Conv-LSTM and Conv-GRU) achieved the highest fidelity in long-term predictions, albeit at the cost of greater computational resources. This broad comparative framework provides practical guidance for selecting appropriate algorithms based on accuracy requirements, interpretability, and hardware constraints.

Our findings underscore the transferability of the proposed modeling pipeline: although trained on data from a single mid-latitude city, the approach requires only local pollen and meteorological inputs and can be applied directly to other regions. Future research should therefore (1) validate generalizability across diverse climatic zones, (2) incorporate automated hyperparameter optimization to further boost model performance, and (3) explore real-time deployment using operational weather forecasts. Ultimately, these efforts will pave the way for user-friendly tools, such as mobile applications to deliver personalized pollen forecasts, support allergists in timing interventions, and help individuals manage exposure effectively.

## Supporting information

**S1 Fig. Graphical abstract.**
(TIF)

**S2 File. Supplementary material.**
(DOCX)

## Acknowledgments

We acknowledge the data providers in the ECA&D project. Klein Tank, A.M.G. and Coauthors, 2002. Daily dataset of 20th-century surface air temperature and precipitation series for the European Climate Assessment. Int. J. of Climatol., 22, 1441-1453. Data and metadata available at https://www.ecad.eu.

## Author contributions

**Conceptualization:** Daniel Bulanda, Małgorzata Bulanda, Małgorzata Sacha, Dorota Myszkowska.

**Data curation:** Daniel Bulanda, Dorota Myszkowska.

**Formal analysis:** Daniel Bulanda, Adrian Horzyk, Dorota Myszkowska.

**Funding acquisition:** Małgorzata Bulanda.

**Investigation:** Daniel Bulanda, Małgorzata Bulanda, Małgorzata Sacha.

**Methodology:** Daniel Bulanda, Adrian Horzyk, Dorota Myszkowska.

**Project administration:** Daniel Bulanda.

**Resources:** Daniel Bulanda, Dorota Myszkowska.

**Software:** Daniel Bulanda.

**Supervision:** Daniel Bulanda, Małgorzata Bulanda, Małgorzata Sacha, Adrian Horzyk, Dorota Myszkowska.

**Validation:** Daniel Bulanda, Małgorzata Bulanda, Małgorzata Sacha, Adrian Horzyk, Dorota Myszkowska.

**Visualization:** Daniel Bulanda.

**Writing – original draft:** Daniel Bulanda, Małgorzata Bulanda, Małgorzata Sacha.

**Writing – review & editing:** Daniel Bulanda, Małgorzata Bulanda, Adrian Horzyk, Dorota Myszkowska.

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
