## [Decision Letter · Decision Letter 0]

2 May 2025

PONE-D-25-12434Comparison of machine learning methods in forecasting and characterizing the birch and grasses pollen seasonPLOS ONE

Dear Dr. Bulanda,

Thank you for submitting your manuscript to PLOS ONE. After careful consideration, we feel that it has merit but does not fully meet PLOS ONE’s publication criteria as it currently stands. Therefore, we invite you to submit a revised version of the manuscript that addresses the points raised during the review process.

We look forward to receiving your revised manuscript.

Kind regards,

Manlio Milanese

Academic Editor

PLOS ONE

3. Thank you for uploading your study's underlying data set. Unfortunately, the repository you have noted in your Data Availability statement does not qualify as an acceptable data repository according to PLOS's standards.

Additional Editor Comments:

This manuscript employs various machine learning techniques to predict and analyze the pollen seasons of birch and grasses. While the manuscript is well developed from a methodological point of view, there are several comments to address before it can be considered for publication. The background and objectives need to be better described.

Please, answer point by point to reviewers comments.verthe results obtained and the main findings of this work.

Reviewers' comments:

Reviewer's Responses to Questions

**Comments to the Author**

1. Is the manuscript technically sound, and do the data support the conclusions?

Reviewer #1: Partly

Reviewer #2: Partly

2. Has the statistical analysis been performed appropriately and rigorously?

Reviewer #1: N/A

Reviewer #2: Yes

3. Have the authors made all data underlying the findings in their manuscript fully available?

Reviewer #1: Yes

Reviewer #2: No

4. Is the manuscript presented in an intelligible fashion and written in standard English?

Reviewer #1: Yes

Reviewer #2: Yes

5. Review Comments to the Author

Reviewer #1: This manuscript employs various machine learning techniques to predict and analyze the pollen seasons of birch and grasses. While the manuscript has the potential to enhance research in this area, there are several minor comments to address before it can be considered for publication

MINOR COMMENTS

1. While meteorological data is included, there is no mention of potential measurement errors or quality variations, which might affect the model's precision. The methodology section could benefit from greater clarity and detail. For example, the description of data preprocessing techniques and modeling strategies could be expanded to include more information on the specific parameters used. In particular, the Authors could to clarify: - The distance between the weather station and the pollen sampling point (over 11 km) could create discrepancies between collected data and actual conditions relevant to the study area.

2. The model relies solely on data from Krakow over a span of 34 years, limiting its applicability to regions with different climatic conditions. It could be useful to comment about that in section weakness or discussion for example.

3. Machine Learning Models: The "default settings" approach for model parameters may restrict performance. Exploring diverse hyperparameter settings would have been beneficial for optimization. In particular, the manuscript does not discuss in detail the impact of data quality (e.g., measurement errors) or the necessity of hyperparameter optimization, which could improve the models' performance.

4. Variable Interpretation: - The feature importance analysis appears detailed but could be expanded to include overlooked or underestimated influential variables

5. Clinical Goal vs Results: Despite the clinical focus on predicting pollen concentrations for personalized treatment, the manuscript does not assess the practical impact of these forecasts on patient health outcomes. It could be useful to comment about that in section weakness or discussion for example.

6. Temporal Overlap: A clear explanation is missing regarding how previous pollen and weather data are synchronized for predictions. Potential temporal overlaps between variables may introduce bias into the models. It could be useful to comment about that in section weakness or discussion for example.

7. Influence of Meteorological Factors: In some parts of the manuscript, it is stated that temperature is the most important environmental factor influencing pollen concentration. However, in other sections, it is suggested that humidity and cloud cover may be of lesser importance. This contradiction could be resolved by providing a clearer and more consistent explanation.

Reviewer #2: Comparison of machine learning methods in forecasting and characterizing the birch and grasses pollen season

General comments

The paper is well developed from a methodological point of view. However, I feel that the manuscript is not sometimes expressed in an aerobiological precise language. Also, the authors should expose correctly the background of this study and explain better the the objectives, the results obtained and the main findings of this work.

- The introduction is very reduced and exposed in very general lines. I think the authors could analyse a brief background of the models of machine learning used in aerobiological studies to predict pollen concentrations, main advantages in comparison to traditional statistical methods and accuracy achieved in the forecasting. Below several interesting and recent related studies using machine learning methods or combination of these methods which is now in practise. After this background, the discussion section should incorporate the main novelty and the most important findings of this study.

- The authors show how pollen time series and meteorological series follow a normal distribution which is strange in the aerobiological field due to the high frequency of zeros and low pollen concentrations during the year. Could you explain this? Anyway, normality assumptions would be only necessary in the case of linear models, are machine learning methods sensitive to the non-normal distribution of data? Discuss this aspect. Perhaps an advantage for using this type of forecasting methods.

- Check the aerobiological terminology of the entire manuscript following standardised terminology in this scientific field (Galán et al., 2017).

Galán, C., Ariatti, A., Bonini, M., Clot, B., Crouzy, B., Dahl, A., Fernandez-González, D., Frenguelli, G., Gehrig, R., Isard, S., Levetin, E., Li, D.W., Mandrioli, P., Rogers, C.A., Thibaudon, M., Sauliene, I., Skjoth, C., Smith, M., Sofiev, M., 2017. Recommended terminology for aerobiological studies. Aerobiologia 33, 293–295. https://doi.org/10.1007/s10453-017-9496-0

- In my opinion, the experiment 2 is not well explained. Indicate better the objectives for each experiment followed.

- The results in the current form is a simple description of figures and tables. In my opinion, the result sections should be a more elaborated presentation of the most important findings of the models.

- Only as a suggestion. It would be interesting to generate a comparative table based on the advantages and disadvantages of the different machine learning methods used in this work. It is a good and useful result for new similar studies.

- Conclusion section should remark the most important findings and the advances in the knowledge, but in the current form, part of the conclusions are general aspects already previously indicated.

Specific comments

- Abstract: "pollens": The word "pollen" has both a singular and plural sense.

- Abstract: "among others": What about the rest of meteorological parameters? Are not relevant?

- Materials and methods (line 30): It is ambiguous. I guess this is the extension of the municipality area.

- Materials and methods (line 48): In "isuploaded" a space is required.

- Materials and methods (Figure 1): Figure 1 is not informative due to the long time series. Perhaps, it would be more useful representing the daily average during the entire series adding lines for the first and third daily quartile to show the variability, or a similar graph.

- Materials and methods (Table 1): Replace "seeds/m3" by "pollen grains/m3".

- Materials and methods (line 79-80): What type of meteorological forecast are used to predict pollen in the future? This is relevant.

- Materials and methods (line 113-120): Replace "seeds/m3" by "pollen grains/m3".

- Materials and methods (line 113-120): What is the meaning of the percentages of each pollen threshold? Is it related to the calculation of the threshold explained in lines 108-110.

- Materials and methods (lines 206-207): Is linear regression the unique method requiring parametric assumptions?

- Materials and methods (line 267): I think "appropriate pollen concentration categories" is not the best statistical term.

- Results: Caption of Figure 5. "Betula" in italics.

- Results (lines 302-303): Why the specific results from the XGBoost are remarked and detailed, and not for the rest of the methods used?

- Results: Caption of Figure 6. "Betula" in italics.

- Results (lines 355-356): A reference is required.

Related interesting literature

Astray et al., 2025. Machine Learning to Forecast Airborne Parietaria Pollen in the North-West of the Iberian Peninsula. Sustainability 17, 1528. https://doi.org/10.3390/su17041528

Cordero et al., 2021. Predicting the Olea pollen concentration with a machine learning algorithm ensemble. Int J Biometeorol 65, 541–554. https://doi.org/10.1007/s00484-020-02047-z

Shokouhi et al., 2024. Spatiotemporal modelling of airborne birch and grass pollen concentration across Switzerland: A comparison of statistical, machine learning and ensemble methods. Environmental Research 263, 119999. https://doi.org/10.1016/j.envres.2024.119999

Zewdie et al., 2019. Applying machine learning to forecast daily Ambrosia pollen using environmental and NEXRAD parameters. Environ Monit Assess 191, 261. https://doi.org/10.1007/s10661-019-7428-x

6. PLOS authors have the option to publish the peer review history of their article (what does this mean?). If published, this will include your full peer review and any attached files.

Reviewer #1: No

Reviewer #2: No

---

## [Author Response · Author response to Decision Letter 1]

29 May 2025

Response to Reviewers

Manuscript title: Comparison of machine learning methods in forecasting and characterizing the birch and grasses pollen season

Manuscript ID: PONE-D-25-12434

Journal: PLOS ONE

Dear Editor and Reviewers,

We would like to thank the Academic Editor and both reviewers for their constructive feedback on our manuscript. We appreciate the time and effort invested in the review process, and we have revised the manuscript accordingly to address all points raised.

Below, we provide a detailed point-by-point response to the reviewers’ and editor's comments. All changes have been incorporated into the revised version of the manuscript, which is submitted along with a tracked-changes file for reference.

General Editorial Requirements

We have revised the manuscript to comply with PLOS ONE style and formatting guidelines. In particular, the Supporting information section has been moved to the end of the manuscript and the description of the corresponding author has been corrected.

We have deposited all necessary author-generated code to GitHub (https://github.com/bionetlabs/PollenForecasting) to ensure unrestricted public access. Appropriate information has been placed in the Materials and methods section.

3. Thank you for uploading your study's underlying data set. Unfortunately, the repository you have noted in your Data Availability statement does not qualify as an acceptable data repository according to PLOS's standards.

The whole material, including row data, pollen data, meteorological data and all the obtained results were stored as project documentation by the Jagiellonian University Medical College.

The meteorological data were collected from the European Climate Assessment and Dataset website [https://www.ecad.eu] as blended data in ASCII file format. All observations were obtained from the Krakow-Balice station. The direct link to the data Policy is as follows: https://knmi-ecad-assets-prd.s3.amazonaws.com/documents/ECAD_datapolicy.pdf.

We acknowledge the data providers in the ECA&D project.

Klein Tank, A.M.G. and Coauthors, 2002. Daily dataset of 20th-century surface air temperature and precipitation series for the European Climate Assessment. Int. J. of Climatol., 22, 1441-1453.

Moreover, to provide a minimal data set necessary to replicate our study, two csv files with Betula pollen and two files with Poaceae pollen data obtained in 2022 and 2023, and the meteorological data of ten selected factors were uploaded to the Open Science Framework with DOI number: DOI 10.17605/OSF.IO/9YZCF and the direct link: https://osf.io/9yzcf/?view_only=d4ccc99d460c46bfa775d72ba6ba3a1b. Appropriate information has been placed in the Materials and methods section.

We have uploaded all main-text figure files to the PACE diagnostic tool and confirmed that they now comply with PLOS figure requirements.

Academic Editor Comments

1. The background and objectives need to be better described.

We have significantly revised the Introduction to better describe the background, the motivations for applying machine learning to aerobiological forecasting, and the gap in current literature. Specific objectives of the study are now clearly defined at the end of the Materials and Methods section.

Reviewer #1 Comments

1. While meteorological data is included, there is no mention of potential measurement errors or quality variations, which might affect the model's precision. The methodology section could benefit from greater clarity and detail. For example, the description of data preprocessing techniques and modeling strategies could be expanded to include more information on the specific parameters used. In particular, the Authors could to clarify: - The distance between the weather station and the pollen sampling point (over 11 km) could create discrepancies between collected data and actual conditions relevant to the study area.

Thank you for pointed out the problem related to the distance between the meteorological and aerobiological stations. We are aware, that the microclimate conditions can differ between both stations, but the main reasons of taking into account the meteorological data from Balice station was that these data are commonly available. It is important not only during the models preparation, but also in the nearest future, when the models will be implemented into the aerobiological data base to predict the pollen concentrations up to date.

For our best knowledge, based on the reports on the meteorological data in both stations (Balice and Kraków city center) the differences between them are relatively low. According to Matuszko et al. (2015) and Matuszko and Piotrowicz (2015), in 2001-2010 annual temperature in Kraków center was higher by 0.7 °C in comparison to Balice station, whereby the differences were observed mainly in winter, while the pollen seasons are finished. The other data, like sunshine, cloud cover, annual relative humidity (77% vs 78%, Kraków and Balice, respectively), were slightly different.

We now discuss the potential impact of measurement errors and spatial discrepancy between the meteorological station and pollen trap in the Materials and Methods under Data subsection. In addition, we have added detailed descriptions of the experimental design and validation procedures to the Materials and Methods section.

Matuszko D., Piotrowicz K., Kowanetz L., 2015, Klimat, [w:] Środowisko przyrodnicze Krakowa. Zasoby, Ochrona, Kształtowanie, Baścik M., Degórska B. (red.), Instytut Geografii i Gospodarki Przestrzennej UJ, Kraków, 81-108.

Matuszko D., Piotrowicz K., 2015, Cechy klimatu miasta a klimat Krakowa, [w:] Miasto w badaniach geografów, Trzepacz P., Więcław-Michniewska J., Brzosko-Sermak A., Kołoś A. (red.), T. 1, Instytut Geografii i Gospodarki Przestrzennej UJ., Kraków, 221-241.

2. The model relies solely on data from Krakow over a span of 34 years, limiting its applicability to regions with different climatic conditions. It could be useful to comment about that in section weakness or discussion for example.

This limitation is acknowledged and discussed in the Discussion section.

3. Machine Learning Models: The "default settings" approach for model parameters may restrict performance. Exploring diverse hyperparameter settings would have been beneficial for optimization. In particular, the manuscript does not discuss in detail the impact of data quality (e.g., measurement errors) or the necessity of hyperparameter optimization, which could improve the models' performance.

We appreciate the Reviewer’s 5 insightful comment. Indeed, exploring the full hyperparameter space could further improve model performance. However, given the large number of models and variants (forecast horizons, taxa), we adopted a standardized pipeline with default parameters to ensure consistency and comparability across all methods.

To address the reviewer’s suggestion, we have now added a discussion in the manuscript highlighting this limitation and acknowledging that more exhaustive hyperparameter tuning could yield better results in some cases. Nonetheless, the models already achieve high performance (e.g., up to 92.2% accuracy), which we believe is sufficient to demonstrate their practical applicability in the context of daily pollen forecasting.

Moreover, regarding data quality, we ensured that the input dataset was thoroughly preprocessed following standardized aerobiological protocols, and we employed robust methods such as moving averages and mutual information analysis to mitigate the effect of noise. This has been clarified in the revised discussion section.

4. Variable Interpretation: - The feature importance analysis appears detailed but could be expanded to include overlooked or underestimated influential variables

We appreciate the reviewer’s suggestion to further examine secondary predictors. Accordingly, we analyzed relative normalized mutual information for the lowest‐ranking meteorological variables: humidity, sunshine duration, radiation, cloud cover, wind speed, sea level pressure, and snow depth. Although these factors fall below the primary drivers in overall rank, they nonetheless contribute meaningfully to model performance. A new paragraph in the Results section now summarizes these insights.

5. Clinical Goal vs Results: Despite the clinical focus on predicting pollen concentrations for personalized treatment, the manuscript does not assess the practical impact of these forecasts on patient health outcomes. It could be useful to comment about that in section weakness or discussion for example.

Thank you very much for drawing attention to the clinical aspect of our work. The practical implication of predicting pollen concentration is the possibility of intensifying antiallergic treatment and avoid activities outdoors at critical moments of pollen concentration to prevent the exacerbation of clinical symptoms in patients. The first stage of our work was the construction and evaluation of models, which is presented in this publication. In the further stages of the project, we plan to use the presented models to build a mobile application that will predict pollen concentrations and will be made available to patients. An appropriate note on this topic has been added to the Discussion section.

6. Temporal Overlap: A clear explanation is missing regarding how previous pollen and weather data are synchronized for predictions. Potential temporal overlaps between variables may introduce bias into the models. It could be useful to comment about that in section weakness or discussion for example.

We thank the reviewer for highlighting the need to clarify how our input windows avoid using future information. In the revised manuscript, we have discussed the potential for autocorrelation bias in the Discussion.

7. Influence of Meteorological Factors: In some parts of the manuscript, it is stated that temperature is the most important environmental factor influencing pollen concentration. However, in other sections, it is suggested that humidity and cloud cover may be of lesser importance. This contradiction could be resolved by providing a clearer and more consistent explanation.

We have revised the relevant paragraphs to present a more coherent interpretation. In particular, wed have expanded the description of Experiment 2 to clearly define its aims and analytical steps and we reworked the appropriate paragraphs in the Results section.

Reviewer #2 Comments

1. The introduction is very reduced and exposed in very general lines. I think the authors could analyse a brief background of the models of machine learning used in aerobiological studies to predict pollen concentrations, main advantages in comparison to traditional statistical methods and accuracy achieved in the forecasting. Below several interesting and recent related studies using machine learning methods or combination of these methods which is now in practise. After this background, the discussion section should incorporate the main novelty and the most important findings of this study.

After correction, the Introduction includes a concise review of the use of machine learning in aerobiology, with emphasis on prior applications and comparative performance. We have also enriched and emphasized the main novelty and the most important findings of this study in the Discussion section.

2. The authors show how pollen time series and meteorological series follow a normal distribution which is strange in the aerobiological field due to the high frequency of zeros and low pollen concentrations during the year. Could you explain this? Anyway, normality assumptions would be only necessary in the case of linear models, are machine learning methods sensitive to the non-normal distribution of data? Discuss this aspect. Perhaps an advantage for using this type of forecasting methods.

We thank the reviewer for this important observation. To clarify, we first excluded all calendar days outside the defined birch and grass pollen seasons. Specifically, those days before the pollen season began and after it ended, when pollen concentrations are zero. After this seasonal trimming, the remaining daily pollen and meteorological values deviate significantly from a normal distribution, as confirmed by Shapiro–Wilk test. We owe you an apology and a clarification regarding our discussion of data normality in the submitted manuscript. Due to a misunderstanding of the Shapiro–Wilk test output and the associated package documentation, we incorrectly reported that both the pollen concentration and meteorological datasets were normally distributed. In fact, the very low p-values (p ≪ 0.05) from our Shapiro–Wilk tests demonstrate that these data strongly deviate from normality. We regret this error and have revised the manuscript to remove the incorrect statements about normality, instead applying and describing appropriate non-parametric analyses. Thank you for catching this mistake, and we appreciate your understanding as we correct our interpretation.

Furthermore, we agree that strict normality is only required for parametric inference in linear regression. In contrast, the remaining models used in our study are non-parametric and thus do not assume any particular distribution of inputs or residuals.

3. Check the aerobiological terminology of the entire manuscript following standardized terminology in this scientific field (Galán et al., 2017).

Thank you for this comment. The aerobiological terminology was adopted to the recommendations proposed by Galan et al. (2017).

4. In my opinion, the experiment 2 is not well explained. Indicate better the objectives for each experiment followed.

We appreciate this feedback and have expanded the description of Experiment 2 to clearly define its aims and analytical steps. Specifically, Experiment 2 was designed to (1) identify which features drive the predictions of our best-performing model, (2) quantify the strength of association between inputs and pollen outcomes via mutual information, and (3) uncover characteristic combinations of meteorological and pollen variables using association rule mining algorithm based on MAGN. We h

---

## [Decision Letter · Decision Letter 1]

22 Jul 2025

PONE-D-25-12434R1Comparison of machine learning methods in forecasting and characterizing the birch and grasses pollen seasonPLOS ONE

Dear Dr. Bulanda,

Thank you for submitting your manuscript to PLOS ONE. After careful consideration, we feel that it has merit but does not fully meet PLOS ONE’s publication criteria as it currently stands. Therefore, we invite you to submit a revised version of the manuscript that addresses the points raised during the review process.

We look forward to receiving your revised manuscript.

Kind regards,

Manlio Milanese

Academic Editor

PLOS ONE

Journal Requirements:

Additional Editor Comments :

Thank you for submitting your above-mentioned manuscript to Plos One.

It has now been evaluated by our experts and we are pleased to inform you that it is principally acceptable for publication in our journal, subject to minor changes.

To assist you in making your alterations, you will find the reviewers' remarks below.

Reviewers' comments:

Reviewer's Responses to Questions

**Comments to the Author**

1. If the authors have adequately addressed your comments raised in a previous round of review and you feel that this manuscript is now acceptable for publication, you may indicate that here to bypass the “Comments to the Author” section, enter your conflict of interest statement in the “Confidential to Editor” section, and submit your "Accept" recommendation.

Reviewer #1: All comments have been addressed

Reviewer #2: All comments have been addressed

2. Is the manuscript technically sound, and do the data support the conclusions?

Reviewer #1: Yes

Reviewer #2: Yes

3. Has the statistical analysis been performed appropriately and rigorously?

Reviewer #1: I Don't Know

Reviewer #2: Yes

4. Have the authors made all data underlying the findings in their manuscript fully available?

Reviewer #1: Yes

Reviewer #2: Yes

5. Is the manuscript presented in an intelligible fashion and written in standard English?

Reviewer #1: Yes

Reviewer #2: Yes

6. Review Comments to the Author

Reviewer #1: Dear Editor and Authors,

I have carefully examined the authors’ point‐by‐point responses to our minor comments and find that they have addressed each concern in a coherent and satisfactory manner:

Measurement error and station‐distance – The authors acknowledge potential microclimate discrepancies between the Balice weather station and the pollen trap and cite two local climatological studies quantifying the differences (≈0.7 °C in annual temperature, ~1 % in humidity). They have now explicitly discussed these limitations and their likely minimal impact in the Materials & Methods.

Geographic generalizability – They have added a clear caveat in the Discussion regarding the single‐city scope and its implications for broader climatic regimes.

Hyperparameter tuning and data quality – The authors explain their rationale for using default settings to maintain comparability across dozens of model‐horizon–taxa combinations. They have also inserted a discussion admitting that more extensive hyperparameter searches might improve performance, and they detail the standardized preprocessing steps (moving averages, mutual information) used to mitigate noise.

Feature‐importance analysis – In response to the suggestion, they carried out and summarized a secondary mutual‐information analysis on the lower‐ranked meteorological predictors, demonstrating that even “minor” factors contribute to overall accuracy.

Clinical implications – They clarify that the forecasting models will underpin a future mobile app for personalized allergy management, and they explain how advance warnings could guide treatment intensification and behavior changes. This note has been added to the Discussion.

Temporal synchronization and bias – The revised manuscript now explicitly states how input windows are constructed to prevent look-ahead bias and addresses potential autocorrelation in the Discussion.

Consistency in meteorological factor interpretation – All relevant paragraphs have been harmonized to present a unified narrative on the primacy of temperature while acknowledging the secondary roles of humidity and cloud cover.

Overall, the authors’ revisions are both thorough and well integrated. They not only justify their methodological choices but also acknowledge remaining limitations, exactly as requested. I believe these responses fully resolve the reviewer’s technical and substantive points.

Reviewer #2: Comparison of machine learning methods in forecasting and characterizing the birch and grasses pollen season

General comments

The authors have considerably improved the manuscript and they have addressed most of my suggestions. I consider that the first round of review has been very positive, but several minor issues should be addressed yet. Figure 5 is a good outcome from my point of view.

- The authors restrict the application of the model to the pollen season, but the pollen season of birch and grasses was not defined. Have they have used any of the common methods to define the pollen seasons? It is relevant as the application of the method influences in the period selected (Tasioulis et al., 2022).

Tasioulis, T., Karatzas, K., Charalampopoulos, A., Damialis, A., Vokou, D., 2022. Five ways to define a pollen season: exploring congruence and disparity in its attributes and their long-term trends. Aerobiologia. https://doi.org/10.1007/s10453-021-09735-2.

- Based on the previous comment, check the clinical method due to the orientation of this manuscript in public health (Pfaar et al., 2017).

Pfaar, O., Bastl, K., Berger, U., Buters, J., Calderon, M.A., Clot, B., Darsow, U., Demoly, P., Durham, S.R., Galán, C., Gehrig, R., Gerth van Wijk, R., Jacobsen, L., Klimek, L., Sofiev, M., Thibaudon, M., Bergmann, K.C., 2017. Defining pollen exposure times for clinical trials of allergen immunotherapy for pollen-induced rhinoconjunctivitis - an EAACI position paper. Allergy 72, 713–722. https://doi.org/10.1111/all.13092

Specific comments

- Abstract: "mean sea level", perhaps you mean "mean pressure at sea level".

- Abstract: "Ambrosia" genus in italics.

- Materials and methods: Replace "pollen counts" by "pollen concentrations".

- Results: "Betula" genus in italics.

- Discussion: In which fields were MAGNs models used previously? Examples.

- Discussion: Replace "grasses pollen" by "grass pollen".

- Conclusion: "mean sea level", perhaps you mean "mean pressure at sea level".

- Discussion: Replace "signWificance" by "significance".

7. PLOS authors have the option to publish the peer review history of their article (what does this mean?). If published, this will include your full peer review and any attached files.

Reviewer #1: **Yes:** Vincenzo Patella

Reviewer #2: No

---

## [Author Response · Author response to Decision Letter 2]

31 Jul 2025

Response to Reviewers

Manuscript title: Comparison of machine learning methods in forecasting and characterizing the birch and grass pollen season

Manuscript ID: PONE-D-25-12434R1

Journal: PLOS ONE

Dear Editor and Reviewers,

We appreciate your constructive feedback and are grateful for the opportunity to revise our manuscript. Thank you for your positive and valuable evaluation.

Below we address each point raised by the reviewers. All changes in the revised manuscript are highlighted in the “Revised Manuscript with Track Changes” file, and the clean version reflects these changes.

Academic Editor Comments

Thank you for your careful evaluation and for confirming that our manuscript is fundamentally suitable for publication pending minor revisions. We appreciate the insightful feedback from you and the reviewers. We will diligently address all comments and submit our revised manuscript by the deadline.

Reviewer #1 Comments

We are very grateful for your thorough re‐evaluation and kind assessment of our revisions. Your confirmation that we have coherently addressed concerns around measurement error and station-distance discrepancies, geographic generalizability, hyperparameter tuning, feature‐importance analysis, clinical implications, temporal synchronization, and consistency in meteorological factor interpretation is incredibly reassuring.

Your detailed recognition of our efforts to quantify microclimate differences, clarify methodological caveats, justify our modeling choices, and integrate secondary analyses means a great deal to us. Thank you for helping us strengthen the manuscript and for your constructive guidance throughout the review process.

Reviewer #2 Comments

We appreciate Reviewer #2’s encouraging feedback and thoughtful recommendations. Each of your points has been addressed in detail below. We are pleased that Figure 5 resonated with you. We’ve also implemented the minor clarifications listed to enhance clarity and precision.

1. The authors restrict the application of the model to the pollen season, but the pollen season of birch and grasses was not defined. Have they have used any of the common methods to define the pollen seasons? It is relevant as the application of the method influences in the period selected (Tasioulis et al., 2022).

Thank you for pointing out the need for greater clarity in our definition of the pollen season. We have rephrased the Materials and methods section accordingly to make explicit that:

- The birch and grass seasons begin on the first day with a pollen concentration > 0 grains/m³ and end on the last day with any detectable pollen, following Dahl et al. (2013).

- All zero‐count days that fall between these start and end dates were retained in the analysis.

Although our method does not employ any of the other widely used pollen‐season calculation algorithms, we elected to omit a detailed comparison of those approaches from the Materials and Methods.

2. Based on the previous comment, check the clinical method due to the orientation of this manuscript in public health (Pfaar et al., 2017).

Thank you for highlighting the importance of the clinical definition of pollen exposure. In response, we have added new sentences in the Discussion section summarizing the EAACI position paper by Pfaar et al. (2017), which recommends daily monitoring of pollen concentrations in immunotherapy trials.

We have added a statement noting that daily pollen monitoring, is crucial for accurately timing and assessing the effectiveness of specific immunotherapy for seasonal allergic rhinoconjunctivitis.

Reviewer #2 Specific Comments

1. Abstract: "mean sea level", perhaps you mean "mean pressure at sea level".

Corrected.

2. Abstract: "Ambrosia" genus in italics.

Corrected in the Introduction section, no such word in Abstract.

3. Materials and methods: Replace "pollen counts" by "pollen concentrations".

Corrected.

4. Results: "Betula" genus in italics.

All occurrences of the word Betula are in italics.

5. Discussion: In which fields were MAGNs models used previously? Examples.

Multi-Associative Graph Networks (MAGNs) have been evaluated across a wide range of benchmarking tasks in the introductory publication (Horzyk A, Bulanda D, Starzyk JA. Construction and Training of Multi-Associative Graph Networks. In: Joint European Conference on Machine Learning and Knowledge Discovery in Databases. Springer; 2023. p. 277–292). The authors evaluated over 70 classification datasets from various fields, including medicine, technology, and biology, demonstrating MAGNs’ broad applicability across disciplines. Beyond benchmarking, MAGNs have also been applied in real-world, domain-specific projects. In the automotive industry, MAGNs have been used as recommendation engine and pattern mining framework at GrapeUp Ltd. In the medical domain, they are applied to the classification and analysis of electrocardiogram (ECG) signals in collaboration with Prof. M. Jastrzębski (Jagiellonian University) and Prof. J.A. Starzyk (Ohio University), with the goal of improving diagnostic accuracy and classification mechanisms. In the field of psychiatry, MAGNs are being implemented within the framework of the project MENTALIO – a decision-support system for diagnostics and therapy in adolescent mental health based on artificial intelligence algorithms, funded by the Polish Medical Research Agency (ABM/2022/7). The project is carried out in collaboration with Prof. M. Pilecki (Jagiellonian University) and the Nivalit company. It focuses on modeling complex psychological and behavioral data to predict children's and adolescents’ behaviors and to support suicide prevention. Notably, MENTALIO received the highest ranking in the national ABM competition for innovative AI-based medical technologies, with total funding of 3,768,206 PLN. These practical implementations further underscore the flexibility of MAGNs in handling diverse, high-dimensional, and interrelated data structures across a wide range of application areas.

6. Discussion: Replace "grasses pollen" by "grass pollen".

Corrected.

7. Conclusion: "mean sea level", perhaps you mean "mean pressure at sea level".

Corrected.

8. Discussion: Replace "signWificance" by "significance".

Corrected.

We trust that these revisions address all the remaining concerns. Thank you again for your valuable feedback. We look forward to your decision.

Sincerely,

Daniel Bulanda

AGH University of Krakow

daniel@bulanda.net

---

## [Decision Letter · Decision Letter 2]

11 Dec 2025

PONE-D-25-12434R2Comparison of machine learning methods in forecasting and characterizing the birch and grass pollen seasonPLOS One

Dear Dr. Bulanda,

Thank you for submitting your manuscript to PLOS ONE. After careful consideration, we feel that it has merit but does not fully meet PLOS ONE’s publication criteria as it currently stands. Therefore, we invite you to submit a revised version of the manuscript that addresses the points raised during the review process.

We look forward to receiving your revised manuscript.

Kind regards,

Rafael Duarte Coelho dos Santos, Ph.D.

Academic Editor

PLOS One

Journal Requirements:

Additional Editor Comments:

Please note the additional information sent by the editors and resubmit the new version.

Reviewers' comments:

Reviewer's Responses to Questions

**Comments to the Author**

1. If the authors have adequately addressed your comments raised in a previous round of review and you feel that this manuscript is now acceptable for publication, you may indicate that here to bypass the “Comments to the Author” section, enter your conflict of interest statement in the “Confidential to Editor” section, and submit your "Accept" recommendation.

Reviewer #2: All comments have been addressed

2. Is the manuscript technically sound, and do the data support the conclusions?

Reviewer #2: Yes

3. Has the statistical analysis been performed appropriately and rigorously?

Reviewer #2: Yes

4. Have the authors made all data underlying the findings in their manuscript fully available?

Reviewer #2: Yes

5. Is the manuscript presented in an intelligible fashion and written in standard English?

Reviewer #2: Yes

6. Review Comments to the Author

Reviewer #2: Comparison of machine learning methods in forecasting and characterizing the birch and grasses pollen season

The authors have addressed all of my suggestions. I consider that this manuscript is ready to be published in a very high scientific quality.

7. PLOS authors have the option to publish the peer review history of their article (what does this mean?). If published, this will include your full peer review and any attached files.

Reviewer #2: No

---

## [Author Response · Author response to Decision Letter 3]

24 Dec 2025

Response to Reviewers

Manuscript title: Comparison of machine learning methods in forecasting and characterizing the birch and grass pollen season

Manuscript ID: PONE-D-25-12434R1

Journal: PLOS ONE

We thank the Academic Editor for the thorough and constructive evaluation of our manuscript from the machine learning perspective. We appreciate the opportunity to improve the clarity, balance, and interpretability of our work. Below we respond point-by-point and describe all corresponding revisions implemented in the revised manuscript.

Academic Editor Comments

Below is a list of individual comments along with the authors' responses.

1. There is an emphasis on the MAGN method, and more space is dedicated to explain it when compared with the other methods. I understand that it is a recently-developed method therefore most people won't know about it, but even so the difference on the details on it and the other methods is noteworthy. One of the authors of this paper is also an author of another MAGN paper used as reference to this one.

We appreciate the Editor’s observation regarding the relative emphasis placed on the MAGN method. Our original intention in providing a more detailed description of MAGN was to ensure that readers unfamiliar with this recently developed approach could understand its structure and rationale, since unlike boosted trees or neural networks, it does not yet have widely available canonical references or textbook-style explanations. MAGN is conceptually different from the other methods evaluated and requires additional clarification to ensure transparency and reproducibility.

Importantly, our goal was not to privilege MAGN but rather to prevent ambiguity in the description of a method that most readers would be encountering for the first time. We also acknowledge that two of the co-authors previously contributed to the development of MAGN, which further motivated us to provide an especially clear and self-contained explanation so that the method could be independently assessed.

To address the concern regarding imbalance, in the revised manuscript we have added one additional paragraph to both the linear regression and k-NN method descriptions to provide clearer methodological balance, as these two models were previously underrepresented relative to the others. This ensures that all methods are presented with a more comparable level of detail while still providing the clarity necessary for readers to understand and replicate the MAGN approach.

2. Figure 2, that describes MAGN, is not really useful, more details are needed. The text that refers to the figure is also not so clear. MAGN shows best or second-best results in mean accuracy, mean MAE, mean execution time and mean memory consumption. From the results it seems very impressive, but the lack of details on the method may hinder its adoption by other researchers.

We thank the reviewer for pointing out that Figure 2 and its accompanying description lacked sufficient detail. In the revised manuscript, we have substantially improved the caption of Figure 2 to clearly explain the roles of sensory fields, sensory neurons, object neurons, duplicate counters, defining connections, and similarity connections within the MAGN architecture. In addition, we have added a new explanatory paragraph in the text that walks the reader through the structure step-by-step and clarifies how these components interact during learning and inference. These revisions enhance the clarity and interpretability of the figure and provide a more accessible explanation for researchers who may not yet be familiar with MAGN.

3. Table 3 shows the source code file name for each method, with references to the source code (e.g. 35, 36, 38) that are not really useful. Why not use a reference to the canonical paper or book that describes the method? Links to the source code could be provided as additional references or footnotes.

We thank the reviewer for this helpful suggestion. In the revised manuscript, we have added canonical theoretical references for each machine learning method listed in Table 3, ensuring that readers can easily locate the foundational literature describing each model. We have also included a link to the MAGN implementation, consistent with the other methods, and extended the table caption to clarify the purpose of the theoretical references and implementation links. These changes improve both the scientific rigor and the reproducibility of the presented methods.

4. Explanation for figure 7 could be more clear: not only comment on the graphical representation of the plot but also point to points of interest on it. In other words, which conclusions are supported by that chart's features?

We thank the reviewer for noting that the interpretation of Figure 7 required greater clarity. In the revised manuscript, we have expanded the description of Figure 7 to go beyond the graphical explanation and now explicitly highlight the most important patterns visible in the plot. We identify which meteorological combinations lead to high-confidence and high-lift rules, describe the differences between Betula and Poaceae, and explain how the size, position, and color of the points support our conclusions. This updated interpretation makes the chart more informative and demonstrates how the association rules contribute to understanding the meteorological drivers of pollen concentration.

5. Figure 6 shows that pollen concentration is the main predictor for pollen concentration... this happens for seasonal predictions. The most important applications are related to prediction of when something will start and stop, which I expected to be the case for this paper (see introduction, line 8). What would happen if we didn't use pollen concentration as an input variable?.

We appreciate the reviewer’s insightful comment regarding the use of past pollen concentrations as predictors and the relevance of forecasting the beginning and end of the pollen season. We agree that onset and offset prediction is an important application. However, the scope of the present study is limited to within-season forecasting and early-season characterization, as stated in line 8 of the introduction. The phrase “early characterization of the upcoming pollen seasons” refers specifically to modeling pollen behaviour after the first measurable occurrence, not to predicting the initial onset itself.

For this reason, experiments excluding pollen concentration as an input variable, while highly valuable for onset/offset prediction, fall outside the objectives of this work. Nevertheless, we acknowledge the importance of the reviewer’s suggestion. Our team is currently developing a follow-up study focused explicitly on pollen season start and end prediction and on evaluating models without historical pollen inputs. We expect this new study to be completed within this year and believe it will address the reviewer’s points in full.

To improve clarity, we revised the sentence to explicitly state that our work focuses on the prediction and characterization of pollen levels after the season has begun. This modification aligns the introduction with the scope of the study and prevents misunderstanding regarding onset prediction.

6. About the title: "Comparison of machine learning methods in forecasting and characterizing the birch and grass pollen season" -- I didn't see much of "characterizing of the seasons", as I expected from Experiment 2, which deals with "characterize the relationships between the input and target variables" but without a good, interpretable explanation -- just a description on lines 360-362, which seems to leave the task to the reader.

Overall, considering the ML methods and applications, I think clearer explanations would help. The problem the authors want to solve is clear (predict pollen concentration), the variables are well explained, but the paper title ("Comparison of machine learning methods in forecasting and characterizing the birch and grass pollen season") seems too broad and results could be better explained.

We appreciate the reviewer’s observation regarding the “characterizing” component referenced in the title. In the revised manuscript, we have expanded the interpretation of Experiment 2 to provide a clearer and more explicit characterization of the relationships between meteorological variables and pollen concentration classes. Additional text now highlights the most informative association rules, explains how these rules reveal underlying seasonal patterns, and connects these findings to established aerobiological knowledge. These enhancements ensure that the characterization aspect of the study is more transparent and accessible to the reader.

We acknowledge that the original presentation placed more emphasis on forecasting results, which may have contributed to the perception that the title was broader than the content. With the strengthened explanations, the paper now more fully reflects both elements, forecasting and characterizing, as indicated in the title.

7. Access to the whole dataset is somehow restricted, rules and contacts to getting to it are listed in the documents (but not on the paper?), I sort of expected a better explanation on why the full data is not available, but I also think this is not an issue that would hinder the publication of the work since there are links to a smaller version of the dataset that allows for replicability.

We thank the reviewer for raising this point. The long-term pollen monitoring data are owned and curated by the Department of Clinical and Environmental Allergology at the Jagiellonian University Medical College, and due to institutional data-governance policies and confidentiality considerations, cannot be made openly available in their entirety. Access requires a formal request, a brief research plan, and confirmation of compliance with data-use and confidentiality rules. To the best of our knowledge, the Data Availability section is the proper and expected location for providing these explanations.

To ensure reproducibility despite these restrictions, we have provided a publicly accessible minimal dataset (including 2022-2023 data points). This subset is sufficient to replicate all analyses and modeling procedures presented in the paper.

8. There are still minor issues with the format of the text, notably in tables.

We thank the reviewer for noting the remaining formatting issues. In the revised manuscript, we have reviewed and corrected all identified formatting problems in the tables and surrounding text.

We hope these revisions resolve the remaining issues. Thank you for your thoughtful feedback. We look forward to your decision.

Sincerely,

AGH University of Krakow

daniel@bulanda.net

---

## [Editor Report · Decision Letter 3]

7 Jan 2026

Comparison of machine learning methods in forecasting and characterizing the birch and grass pollen season

PONE-D-25-12434R3

Dear Dr. Bulanda,

We’re pleased to inform you that your manuscript has been judged scientifically suitable for publication and will be formally accepted for publication once it meets all outstanding technical requirements.

Kind regards,

Rafael Duarte Coelho dos Santos, Ph.D.

Academic Editor

PLOS One

Additional Editor Comments (optional):

Thanks for replying to the reviewers' questions!
---

## [Editor Report · Acceptance letter]

PONE-D-25-12434R2

PLOS ONE

Dear Dr. Bulanda,

I'm pleased to inform you that your manuscript has been deemed suitable for publication in PLOS ONE. Congratulations! Your manuscript is now being handed over to our production team.

Kind regards,

on behalf of

Dr. Manlio Milanese

Academic Editor

PLOS ONE